# Soil microbiome perturbation impedes growth of *Bouteloua curtipendula* and increases relative abundance of soil microbial pathogens

**Alisiara Hobbs[1], Daisy Ochoa-Rojas[1], Christine E. Humphrey[2], John A. Kyndt [1]\*, Tyler C. Moore[1]\***

**1** Department of Biology, College of Science and Technology, Bellevue University, Bellevue, Nebraska, United States of America, **2** Department of Bioengineering, School of Engineering, Rice University, Houston, Texas, United States of America

\* jkyndt@bellevue.edu (JAK); tymoore@bellevue.edu (TCM)

## Abstract

*Bouteloua curtipendula* (sideoats grama) is a valuable prairie grass for livestock forage, supporting food webs of herbivorous insects, reducing soil erosion, and limiting weed infiltration in urban grasslands. Efficient establishment of *B. curtipendula* in prairie restorations and urban plantings could drastically improve long-term functionality of the space. Soil microbial communities have been linked to plant germination, growth, and drought tolerance in many plant species, however little is known about the factors contributing to *B. curtipendula* germination and early growth. In this study, we used sterilized soil to examine the impact of soil microbes on *B. curtipendula* growth under greenhouse conditions. We found *Bouteloua curtipendula* emergence and growth to be impaired in sterilized soil compared to not-sterilized soil. Using high throughput sequencing of the soil, we found that *B. curtipendula* grown in sterilized soil induced a greater proportion of plant pathogens and fewer nitrifying bacteria as compared to when grown in not-sterilized soil. For example, there was a significantly higher proportion of *Acidovorax*, *Cellvibrio*, and *Xanthomonas* which are known to contain plant pathogens, while plant-growth promoting bacteria, like *Rhodopseudomonas,* were significantly higher in the not-sterile soil conditions. We found that soil sterilization and growth of *B. curtipendula* changed the relative abundance of metabolic subsystem genes in the soil, however, by seven weeks after seeding, *B. curtipendula* transformed the bacterial community of sterile soil such that it was indiscernible from not-sterile soil. In contrast, fungal communities in sterilized soil were still different from not-sterilized soil seven weeks post-seeding. It appears that the bacteria are involved in the initial establishment of beneficial conditions that set the stage for a robust fungal and plant seedling development.

**Data availability statement:** All of the WGS metagenomic and targeted fungal sequencing datasets were deposited into NCBI Genbank under BioProject PRJNA1163419 and the WGS data are accessible with the following SRA numbers: SRR30751922-SRR30751939 and SRR30754582-SRR30754613. The targeted fungal ITS sequencing files are available with the following SRA numbers: SRR30786940-SRR30786954. The links to the data are provided in the paper.

**Funding:** This work was supported by the Wilson Enhancement Fund for Applied Research at Bellevue University, for which Dr. John A. Kyndt and Dr. Tyler C. Moore were the recipients.

**Competing interests:** The authors have declared that no competing interests exist.

## Introduction

Classic conservation efforts have focused on expansive nature preserves, and in addition, recent studies are acknowledging the opportunity for native grasslands in cities creating so-called 'pocket prairies' [1]. Native grassland pockets within cities support insect diversity [2,3], increase pollinator activity [4–6], and increase biological control of pest organisms [7,8]. Urban habitat spaces dominated by regionally-native plants can also enhance the diversity and breeding success of native birds [2,9,10]. In contrast, invasive plant species reduce the capacity for urban plant communities to support bird and insect biodiversity [11,12]. Thus, the capacity for pocket prairies to promote urban biodiversity depends upon the plant community structure.

Because plant composition is an important indicator of urban pocket prairie function, it is important to understand how management efforts could better support diverse native plant communities. One approach to reducing invasive species infiltration to pocket prairies involves a thick planting of native grasses or sedges at the base of forbs to block weed seed access to the soil, shade the soil, and outcompete new seedlings for water and nutrients. This strategy, referred to as "matrix planting" has been shown to be successful in anecdotal reports from landscape designers [13]. Matrix planting relies on a robust clumping or spreading groundcover to fill in the gaps between ornamental forms in a plant community. One commonly utilized matrix plant is *Bouteloua curtipendula*, chosen for its bunching form, drought tolerance, easy establishment from seed, and low growth form [13]. A recent study found that broadcast seeding of prairie mixes results in more desirable plant species and less invasive plants compared to more labor-intensive methods (such as planting plugs) [14]. However, strategies that promote rapid and stable establishment of *B. curtipendula* and other matrix plants could therefore increase overall functionality of pocket prairies by contributing to native plant diversity and reducing invasion by exotic plant species.

Soil microbial communities are likely an important, yet understudied, aspect of urban pocket prairie functionality. Soil microbes have intrinsic ecosystem functions which could add to the value of pocket prairies, such as carbon sequestration and reduction of pollutants [15–17]. Soil bacteria mineralize organic nitrogen to more bioavailable forms [18], break down water pollutants passing through soil, and sequester carbon to lessen the impacts of atmospheric carbon dioxide on climate change [19–21]. In fact, urban soil likely provides the same ecosystem services as non- urban soil [17], especially in areas revegetated with locally-native plants [22–23]. These soil microbial processes promoted by urban native plant habitats are likely critical for urban agriculture, human quality of life, and supporting biodiversity within cities.

Soil microbial communities may also be an important factor in determining the rate of establishment of matrix plants in pocket prairies. In order to propagate and establish, plants need to germinate from seeds, uptake water and nutrients from the soil, gather sunlight for photosynthesis (in order to fix carbon dioxide into glucose in the Calvin cycle), and survive stressors which perturb these processes. Soil microbes have long been linked to germination and growth of early seedlings, with

both inhibitory and stimulatory effects [24]. Endophytic bacteria have been shown to increase quinoa germination rates by activating immune-related signaling pathways within the seed [25]. A more recent broad scale study used high throughput sequencing to demonstrate a link between soil microbial community and germination across several plant families [26]. Thus, it is possible that microbial community processes could promote the germination of matrix plants such as *Bouteloua curtipendula* and thereby increase the success of pocket prairie plantings. Understanding these microbiome-plant relationships in native grasses, such as *B. curtipentula* could potentially open up possibilities for incorporating target strains into seeds, similar to what has been performed in crop plants [27].

Soil bacteria are also able to promote growth of seedlings and mature plants. One mechanism of soil microbe stimulation of plant growth is through enzymes like 1- aminocyclopropane-1-carboxylate (ACC) deaminase, which reduces plant stress hormones such as ethylene [28]. Soil bacterial metabolites also increase water availability [29], which could promote plant establishment during periods of drought. Soil microbial decomposition could also promote plant growth. One study found fungal endophytic symbionts increased plant litter decomposition and subsequent nutrient availability in the soil [30]. Soil bacteria also increase nitrogen and phosphate availability for plants, which could be important for plant growth when these nutrients are limiting [18,19,31]. More recent work has shown the stoichiometric ratio of fungi to bacteria is important for grassland litter decomposition, with low fungi:bacteria ratios corresponding to high nitrogen immobilization by bacteria [32].

Further evidence for the benefits of soil microbes on prairie plant growth comes from work in remnant prairies and prairie restoration. Three herbaceous legume perennials (*Lespedeza capitata, Amorpha canescens,* and *Dalea purpurea*) had greater plant biomass when inoculated with remnant prairie soil [33]. In addition, remnant prairie inoculation significantly increased the number of leguminous root nodules on *Amorpha canescens* and *Lespedeza capitata* (but not *Dalea purpurea*). However, inoculation did not significantly impact the root colonization into the soil [33]. These results are similar to those showing that inoculation of prairie restorations with remnant prairie soil accelerated succession (defined as an increased abundance of late-successional plant species) [34].

The urban setting of pocket prairies makes them unique from prairie remnants, but there is evidence that microbes could help plants adapt to these novel conditions. For instance, fitness of *Brassica rapa* during experimental drought was largely linked to microbial communities that rapidly adapted to the new conditions [35]. It is likely that soil microorganisms also increase prairie plant tolerance of elevated heat and drought in urban pocket prairies compared to remnant prairies.

Urban pocket prairies also differ from native remnant prairies in the degree of soil disturbance. In mammalian intestines, disruption of the microbial community with antibiotics increases relative abundance of opportunistic pathogens [36]. Perturbation of soil microbial communities could similarly open niches for less beneficial or even pathogenic bacteria. If so, these disturbances could impede the establishment of prairie plants and reduce the likelihood of urban pocket prairie long-term success. Indeed, soil sterilization reduced the above-ground biomass of *Bouteloua gracilis* [37], but it remains unclear which microbial changes were associated with reduced *B. gracilis* growth.

Although the general role of soil microbes in plant germination, nutrient uptake, growth, and stress tolerance has been described, it remains unclear how soil microbial communities could promote the establishment of matrix plants such as *Bouteloua curtipendula* following acute soil perturbations. In this study, we investigate the role of microbial communities on the growth of the matrix plant *B. curtipendula* by comparing plant growth with or without acute perturbation of the soil microbes (by autoclave sterilization). In addition, we use high throughput sequencing to measure the changes to bacteria and fungi in response to soil perturbation and study its impact on the growth of *B. curtipendula* under controlled conditions. These results could have important implications for the efficient establishment of grasses in urban grassland restorations.

## Methods

### Plant growth measurements

*B. curtipendula* was grown from seed under four conditions; sterilized planted, not sterilized planted, sterilized not planted, and not sterilized, not planted. Standard potting soil and *B. curtipendula* seeds (obtained from Prairie Legacy; SKU

BOU01) were used in a 72-cell seed tray. Sterile soils were created through autoclaving the potting soil for two complete cycles at 132 degrees Celsius for 30 minutes at 15 psi. Soil samples of sterile and not sterilized groups were collected at the time of planting (week 0). Planted soil received one seed ¼ cm deep in each well. Each condition had 6 replicates, and each replicate was placed in a controlled greenhouse environment with temperature range from 19−25 degrees Celsius, humidity from 60−70%, light levels from 150−425 PAR, and twice daily watering at 5:00 am and 4:00 pm. Temperature, humidity and light levels were controlled using the Greenhouse Wadsworth Seed control systems and measured using aspirated temperature/humidity sensors and PAR light sensors (Model number PCB:F-9327–2 Rev 5.0; Wadsworth, Arvada, Colorado, US). The height and number of blades were recorded weekly for 7 weeks using a standard ruler. All measurements were performed by a single individual to ensure consistency over time. At the start of the study, measurements were compared between two individuals to conclude minimal impact on between-measurer variability. In order to allow longitudinal study without disturbing the soil between collection points and to pair-match soil samples with plant growth data, germination was inferred as visible cotyledon. Soil samples were collected at weeks 0, 4, and 7. At collection points, the plant material was removed and the entire plug of remaining soil was homogenized and sampled. Roots were separated and dried at week seven to record dry root mass.

## Soil pH measurements and LB agar plating

To measure the soil pH, 500 mg of soil samples were diluted in 5 ml of distilled water and tested with API 5 in 1 test strips (Mars Fishcare North America, PA, USA). Gross morphological analysis of soil microbiota was performed by plating approximately 1 g of soil directly on LB nutrient agar plates and incubating them for one week at room temperature.

## DNA extraction

Soil samples were homogenized, and 250 mg of soil was used for DNA purification using the Qiagen DNeasy Power-Soil Pro Kit according to manufacturer's protocol (ref: 47014). The extracted DNA was EtOH precipitated for additional cleanup. Samples were resuspended in 4 mM TrisHCl, pH 7.0. DNA analysis using QuBit (Invitrogen, Qubit 3.0 Fluorometer, model Q33216) and NanoDrop (Thermo Scientific, model NanoDrop OneC, ND-ONEC-W) showed final concentrations ranging from 5.0 ng/ul to 32.7 ng/ul and A260/280 ratios ranging from 1.6 to 2.0. For whole genome-based sequencing we used 100–500ng of each sample and for fungal ITS amplification samples were diluted to 5.0 ng/ul for the initial amplification reaction.

## Next-generation sequencing

Whole genome metagenomics sequencing libraries were prepared using the Illumina Nextera DNA Flex Library Prep kit and library quality and concentration was assessed using Qubit (Invitrogen, Qubit 3.0 Fluorometer Q33216) and Nanodrop (Thermo Scientific, model NanoDrop OneC, ND-ONEC-W). All libraries had A260/280 ratios higher than 1.8 and concentrations higher than 10 nM. Libraries were pooled at 10 nM as described in the Illumina Nextera DNA Flex Library Prep protocol, and were sequenced by an Illumina MiniSeq using 500µl of a 1.8pM library using a High-output cassette. Paired-end (2x150 bp) sequencing generated between 139,594 and 9,540,606 reads (>Q30), and 21 Mbps-1.4 Gbp of data for each of the samples. One replica sample failed to produce sufficient sequencing data (from the sterile soil planted with *B. curtipendula* and harvested at 4 weeks) and was eliminated from the analysis.

While whole genome sequencing is a shotgun sequencing approach that provides a comprehensive view of the microbiome, including functional genes and potential for strain-level resolution, however when using a WGS approach, the fungal species are often underrepresented, due to the abundance of bacteria and the lack of high quality curated fungal genomes in the databases [38,39]. Therefore, a more targeted approach is beneficial and fungal ITS libraries were prepared by using the Fungal Metagenomic Sequencing protocol from Illumina, which uses a pool of 8 forward and

7 reverse amplicon primers that target the fungal ITS1 region between the 18S and 5.8S rRNA genes. These include the ITS1-F and ITS2 amplicon primers from Bellemain et al. [40], that are widely used for fungal barcoding studies. Amplicon primers were synthesized by Sigma Aldrich (Merck KGaA, Darmstadt, Germany). Targeted ITS amplicon sequencing was also performed using an Illumina MiniSeq using 500µl of a 1.8pM library using a Mid-output cassette. Duplicate samples were sequenced for each condition. This generated between 517,494 and 1,251,248 reads (>Q30) and between 78.1 and 188.9 Mbp of data per sample. Both the WGS metagenomic and targeted fungal sequencing datasets were deposited into NCBI Genbank under BioProject PRJNA1163419 and the WGS data are accessible with the following SRA numbers: SRR30751922-SRR30751939 and SRR30754582- SRR30754613. The targeted fungal ITS sequencing files are available with the following SRA numbers: SRR30786940-SRR30786954.

### Data analysis

Quality control of the raw reads was performed using FASTQC in BaseSpace (Illumina; version 1.0.0). Adapter sequences were removed and reads with low quality scores (average score < 20) were filtered out using the FASTQ Toolkit within BaseSpace (Illumina, version 2.2.0). Taxonomic classification was analyzed using MG- RAST (version 4.0) [41]. After upload to MG-RAST, data was preprocessed by using SolexaQA [42] to trim low-quality regions from FASTQ data. Potential human sequencing reads were removed using Bowtie [43] (a fast, memory-efficient, short read aligner), and only filtered reads passed into the next stage of the annotation pipeline. MG-RAST uses DEseq for normalization [44].

Fungal ITS samples were all processed in BaseSpace, where primer and adapter sequences were removed and reads with low quality scores (average score < 20) were filtered out using the FASTQ Toolkit (Illumina, version 2.2.0). The Metagenomics app within BaseSpace (version 1.0.1) was used to perform a taxonomic classification. The UNITE v9 Fungal database (updated November 2022) was used for this analysis. Default parameters were used for all software unless otherwise noted.

Data from the BaseSpace Metagenomics and MG-RAST output was organized, reformatted, and analyzed using R version 4.1.3. Sequence data were rarefied to the minimum read count per sample using rrarefy() within Vegan version 2.5–7 package in R in order to avoid analysis errors due to unequal reads per sample [45]. For beta diversity, average Bray-Curtis distances were calculated from rarefied data using avgdist() in Vegan sampling the minimum number of reads per sample over 100 iterations. Rarefaction allowed for normalizing to differences in read count between samples in a non-biased manner. To statistically compare the overall bacterial community structure across different soil treatment conditions, permutational multi- variate analysis of community-level variation was performed using adonis() in Vegan with 100 permutations to compare Bray-Curtis distances between sterilized soil and non-sterilized soil as well as between planted and non-planted soil [45]. Shannon diversity index was calculated at the genus taxonomical level for rarefied data using Vegan [45]. Graphs were generated with the ggplot2 R package [46]. The specific R scripts used are provided as supplemental information in S1 File.

### Results

#### Acute soil microbiome perturbation slows growth of B. curtipendula

After one week, *B. curtipendula* sown in sterilized soil visibly germinated in 1/12 samples, while *B. curtipendula* sown in not-sterilized soil visibly germinated in 8/12 samples. By week 2, we observed germination of *B. curtipendula* in 6/12 sterilized soils and 11/12 not sterilized soils (Fig 1A). By week 7, *B. curtipendula* emerged in all samples across all soil types. *B. curtipendula* grown in sterilized soil developed significantly fewer blades of grass compared to *B. curtipendula* grown in not-sterilized soil (P < 0.05 by ANOVA) (Fig 1B). By week 7, *B. curtipendula* grown in not-sterilized soil averaged approximately 7.5 blades of grass per sample. In contrast, *B. curtipendula* grown in sterilized soil averaged approximately 2.5 blades per grass at the same time point (Fig 1B). The cumulative blade length of *B. curtipendula* grown in sterilized soil was also

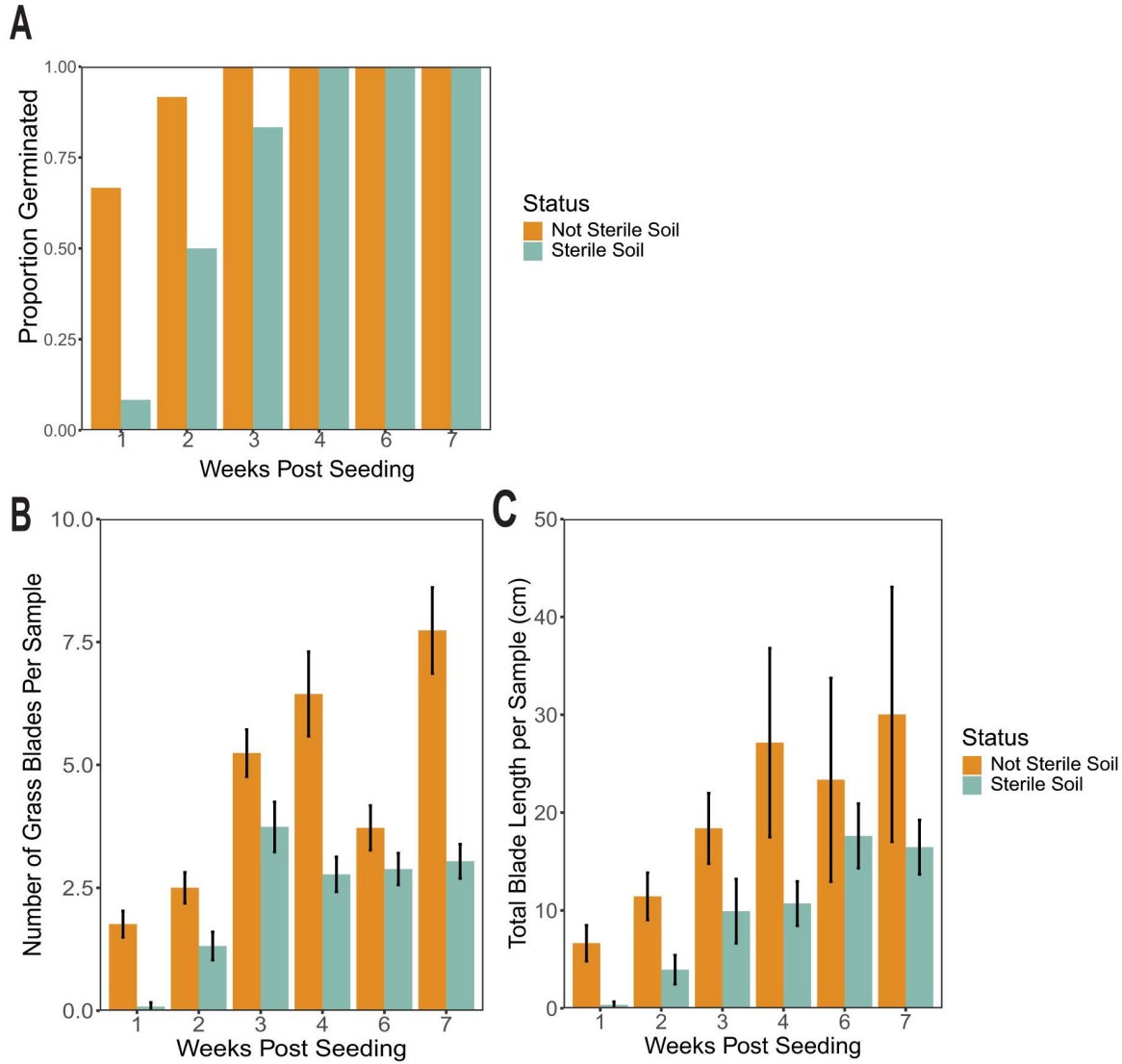

**Fig 1. Acute soil perturbation impedes growth of *Bouteloua curtipendula*.** Potting soil was sterilized via autoclaving (as described in the materials and methods, "Sterile") or left unsterilized ("Not Sterile") and then added to 72 plug trays. *B. curtipendula* was seeded at week 0, and the germination was counted as visible growth of seedlings as a proportion of the total samples **(A)**, the number of grass blades per plug was counted (B) and the cumulative length of all the grass blades was measured **(C)**. Bars represent means of n = 12 samples per group (weeks 1-4) or n = 6 samples per group (weeks 6-7) +/- SEM. Differences between means was compared using a one-way ANOVA. **(A)** Comparison between sterile and not sterile, P < 0.001; comparison across weeks, P < 0.001; interaction between sterility and weeks, P = 0.0138. **(B)** Comparison between sterile and not sterile, P = 0.0006; comparison across weeks, P < 0.001, interaction between sterility and weeks, P = 0.866.

significantly less than that of *B. curtipendula* grown in not-sterilized soil (P < 0.05 by ANOVA) (Fig 1C), but the difference was mostly observed during the first 4 weeks. By week 6 and 7, we could not detect a significant difference in cumulative *B. curtipendula* grass length between the two conditions (P > 0.05 by Tukey post-test for multiple comparisons) (Fig 1C).

In order to determine if reduced growth of *B. curtipendula* following soil sterilization was due to changes in soil pH, we measured the pH of soil at time 0 (with or without sterilization), 4 weeks post-planting, and 7 weeks post-planting. All samples had similar pH values (between 8.1 and 8.7), with no notable differences between groups.

## Response of soil microbiome to acute perturbation and B. curtipendula growth

In order to study the soil microbes associated with differential *B. curtipendula* growth, we used whole genome high throughput short read sequencing (HTS) and targeted amplicon HTS of fungus-specific ITS elements. To study the soil bacterial community, we used the WGS output and selected all reads aligned to the domain Bacteria and performed NMDS analysis at the taxonomic level of genus. As expected, we observed a significant effect on the overall soil bacterial community by soil sterilization, however the impact of planting with *B. curtipendula* was significantly different in the sterile versus the not-sterilized soil (P<0.05, Adonis) (Fig 2A). In the not- sterilized samples, the bacteria community was overall similar in both the soil planted with *B. curtipendula* as in the not-planted soil. This was true at both 4 and 7 weeks post- seeding (Fig 2). However, in the sterilized samples, soil planted with *B. curtipendula* had a unique bacterial community compared to unplanted soil at 4 weeks post-seeding. By 7 weeks post-seeding into the sterilized soil, the bacterial community in both planted and unplanted soils converged toward the bacterial community of not-sterilized soils (Fig 2A). These results indicate a transient disruption to the soil bacterial community resulting from planting *B. curtipendula* into sterilized soil, which did not occur in the not- sterilized soil.

We also observed a significant interaction between soil sterilization and *B. curtipendula* planting on the soil fungal community (P<0.05, Adonis). In contrast to what we observed in bacteria, fungal communities in sterilized soil were similar between planted and unplanted samples at week 4 (Fig 2A). Instead, planting *B. curtipendula* into sterilized soil induced a unique fungal community compared to unplanted sterilized soil which can be observed as a significantly different mycobiome at week 7 (Fig 2B). Interestingly, fungal communities in sterilized soil planted with *B. curtipendula* began to more closely resemble the seed fungal community by week 7, suggesting seed-origin fungi were better able to colonize sterilized soil (Fig 2B).

## Functional diversity changes in response to soil microbiome perturbation and planting with B. curtipendula

At week 0, sterilized soil had drastically less bacterial diversity and richness than not-sterile soil (Fig 3A, 3C). Bacterial diversity and richness of the seed before planting it (week 0) was intermediate between sterile and not-sterile soil (Fig 3A, 3C). We observed slightly more bacterial diversity and richness in sterilized soil planted with *B. curtipendula* compared to unplanted soils at 4 weeks post-seeding (Fig 3A, 3C). However, bacterial richness and diversity were equal across all conditions by 7 weeks post-seeding (Fig 3A, 3C). In contrast, we did not observe a decrease in fungal diversity immediately following sterilization (week 0) (Fig 3B). However, sterilized soil at 4 weeks showed significantly less fungal diversity than not sterilized soil. Although fungal diversity of the sterilized soil increased from 4 weeks to 7 weeks, diversity was not fully restored to levels observed in not sterilized soil. Of note, we observed more fungal diversity and richness at 7 weeks in sterilized soil planted with *B. curtipendula* compared to unplanted soil.

The fact that the comparison between "sterile not planted" and "sterile planted" shows both higher bacterial and fungal diversity can be explained by the fact that the plants likely provided additional niches to support new bacterial and fungal taxa.

Fig 4 shows the metabolic pathway groupings (level 1 subsystem) from whole genome sequencing. Carbohydrates, clustering-based subsystems, amino acids and derivatives, and protein metabolism were the most abundant across all groups. Among the sterilized soil at week 0, carbohydrates and dormancy and sporulation were overrepresented compared to other groups. In contrast, sterilized soil at week 0 had a lower percentage of protein metabolism and respiration subsystems. Aside from the sterilized soil at week 0, the majority of metabolic pathways were similar between groups.

Fig 5A shows the relative abundance (in percent) of selected level 3 subsystems across the treatment groups. Spore germination was overrepresented in sterilized soil at week 0 (Fig 5B), suggesting spore-forming bacteria were the primary species to survive autoclaving. Although underrepresented in sterile soil at week 0, Ton and Tol transport systems (within Membrane Transport) and iron acquisition were overrepresented in sterile planted soil at week 4 compared to all other groups (Fig 5D, I). Cobalt/Zinc/Cadmium resistance was also elevated in sterile planted soil at 4 weeks compared to other

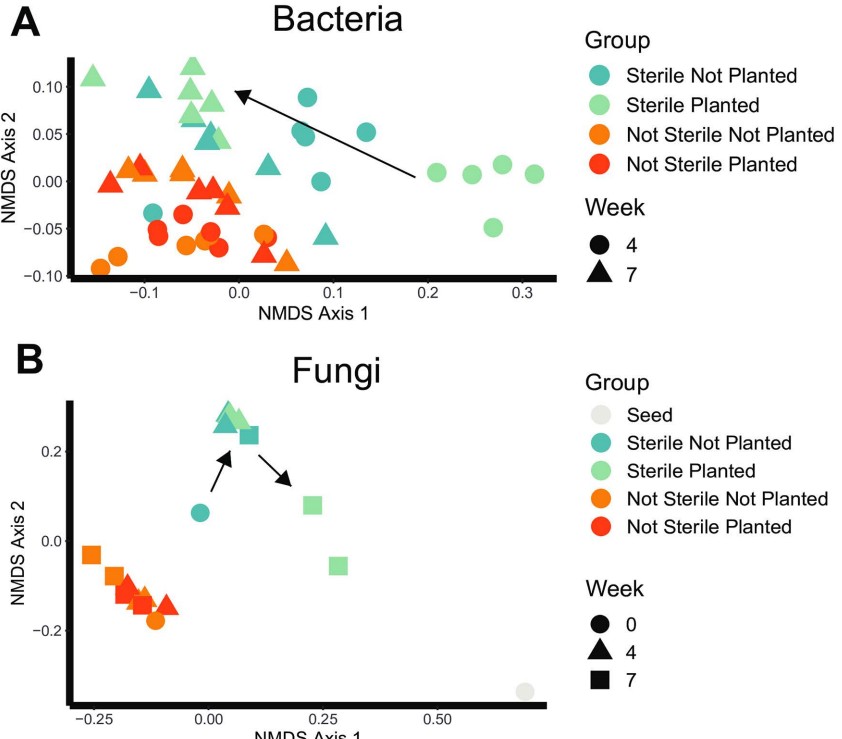

**Fig 2. Soil sterilization and planting *Bouteloua curtipendula* shift the soil microbial communities.** Potting soil was sterilized via autoclaving (as described in the materials and methods, "Sterile") or left unsterilized ("Not Sterile") and then added to 72 plug trays. *B. curtipendula* was seeded at week 0 ("Planted") or left unseeded as a control ("Not Planted"). At week 4 and seven, soil was collected for DNA purification and Illumina sequencing (as described in the materials and methods). The *B. curtipendula* seed ("Seed") and soil samples immediately after autoclaving ("Week 0") were taken as a control. Each symbol represents an individual sample. Multi-ordinate variance was compared by Adonis. **(A)** Comparison between sterile and not sterile, P = 0.001; comparison between planted and not planted, P = 0.001; comparison across weeks, P = 0.001; interaction between sterility and planted/not planted, P = 0.011; interaction between sterility and weeks, P = 0.001; interaction between weeks and planted/not planted, P = 0.001; interaction between sterility, weeks, and planted/not planted, P = 0.001. **(B)** Comparison between sterile and not sterile, P = 0.001; comparison between planted and not planted, P = 0.002; comparison across weeks, P = 0.007; interaction between sterility and planted/not planted, P = 0.201; interaction between sterility and weeks, P = 0.006; interaction between weeks and planted/not planted, P = 0.057; interaction between sterility, weeks, and planted/not planted, P = 0.063. Arrows indicate changes from week 4 to week 7 in the sterilized soil microbial community following planting with *B. curtipendula*.

groups (Fig 5E). In contrast, nitrogen fixation and carbon monoxide- induced hydrogenase was underrepresented in sterile planted soil at 4 weeks compared to other groups (Fig 5G, H). By 7 weeks post-seeding, relative abundance of each subsystem in the sterile planted soil more closely resembled the other groups (Fig 5).

### Impacts of soil microbiome perturbation and planting with B. curtipendula on individual bacterial and fungal taxa

Almost all bacteria detected in the week 0 sterilized soil belonged to the phylum *Firmicutes* (Fig 6A). In contrast, not sterilized soil at week 0 was primarily composed of the phyla *Proteobacteria, Planctomycetes, Bacteroidetes,* and *Actinobacteria* (Figs 6A, 7A, B). At 4 weeks post-seeding in the sterilized soil, Proteobacteria and Bacteroidetes were overrepresented compared to planted not sterilized soil (Figs 6A, 7B). The phylum *Actinobacteria* was less abundant in sterilized planted soil at 4 weeks compared to not sterilized planted soil (Figs 6A, 7A). By 7 weeks post-seeding, the relative abundance of *Bacteroidetes, Actinobacteria*, and *Proteobacteria* were similar between all groups (Figs 6A, 7A, B).

Amongst the fungi, *Ascomycota*, *Basidiomycota*, and *Chytridiomycota* were the most abundant in the soil (Fig 6B). Nearly all fungi on the seed belonged to the phyla *Ascomycota* or *Basidiomycota*, with an approximate 2.8:1 ratio of

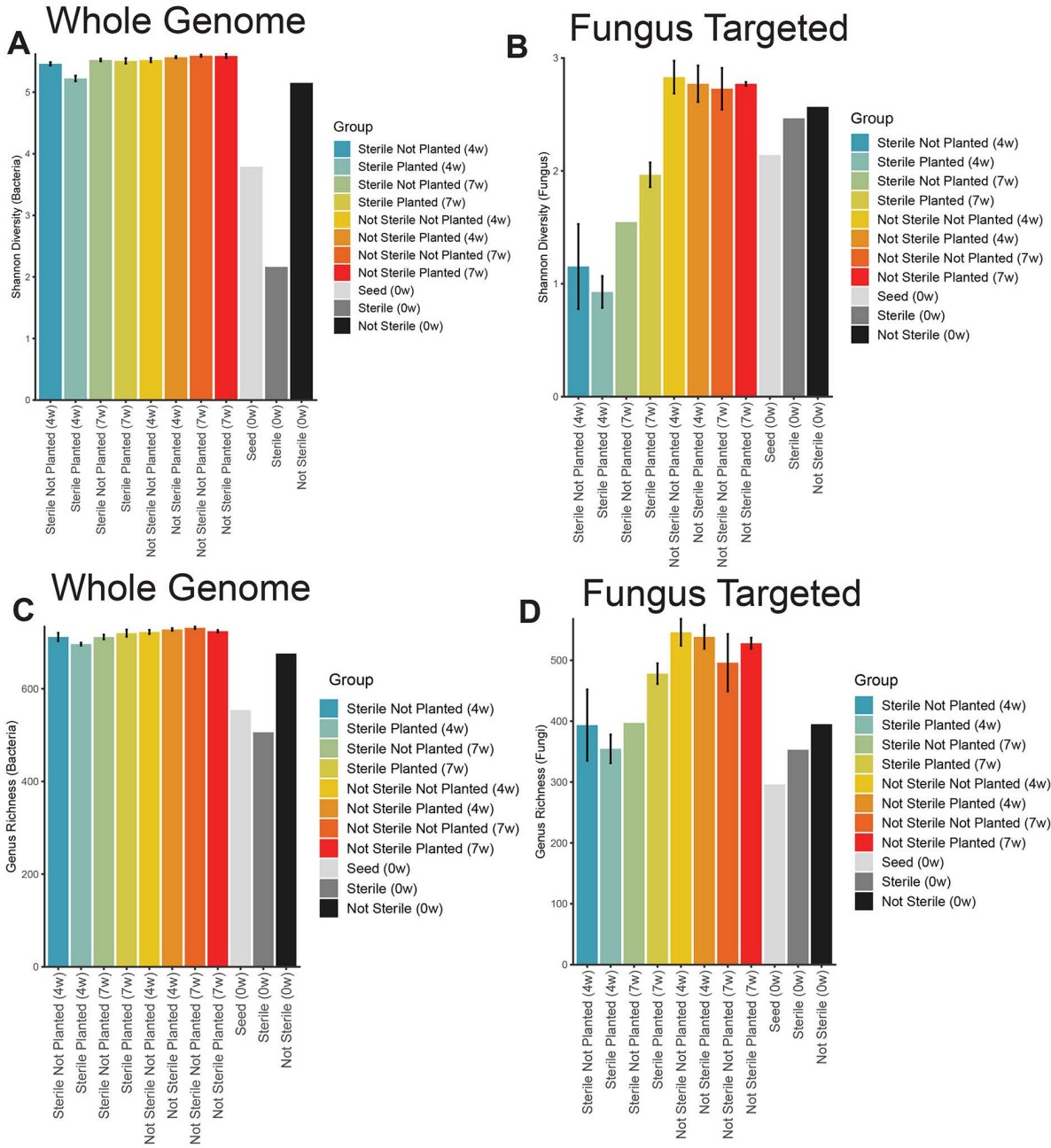

**Fig 3. Soil microbial alpha diversity (Shannon diversity index) and richness changes in response to soil sterilization and *B. curtipendula* growth.** Potting soil was sterilized via autoclaving (as described in the materials and methods, "Sterile") or left unsterilized ("Not Sterile") and then added to 72 plug trays. *B. curtipendula* was seeded at week 0 ("Planted") or left unseeded as a control ("Not Planted"). At week 4 and seven, soil was collected for DNA purification and Illumina sequencing. Bars represent means +/- SEM of alpha diversity **(A, B)** or richness **(C, D)** within whole genome sequencing **(A, C)** or fungus ITS-targeted sequencing **(B, D)**, n=6 samples per group (week 0 and seed, n=1).

*Ascomycota* to *Basidiomycota* ([Figs 6B](), [7C]()-[D]()). The fungal phylum *Ascomycota* was overrepresented in sterilized soil compared to not sterilized soil at both 4 and 7 weeks, representing greater than 75% of all fungal phyla reads ([Figs 6B](), [7C]()). In contrast, the phylum *Basidiomycota* was less abundant in sterilized soils compared to unsterilized soils

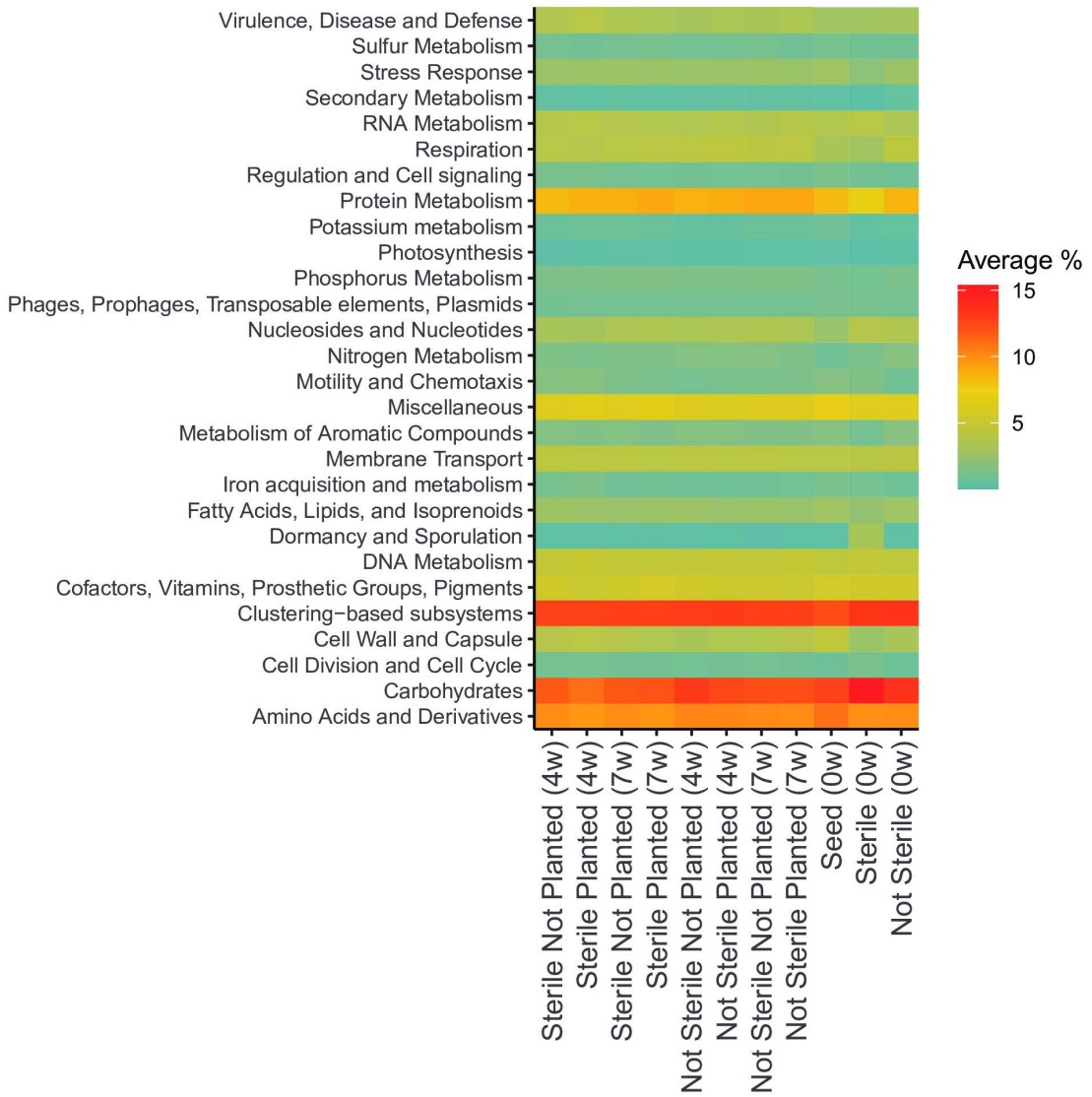

**Fig 4. The impact of soil sterilization and planting with *Bouteloua curtipendula* on soil metabolic pathways.** Potting soil was sterilized via autoclaving (as described in the materials and methods, "Sterile") or left unsterilized ("Not Sterile") and then added to 72 plug trays. *B. curtipendula* was seeded at week 0 ("Planted") or left unseeded as a control ("Not Planted"). At four and 7 weeks, soil was collected for DNA purification and Illumina sequencing. Whole-genome sequencing was mapped to metabolic systems using MG-RAST. Colors represent the average percent of each metabolic system within a group (n = 6; week 0 and seed, n = 1).

(Figs 6B, 7D). Planting sterilized soil with *B. curtipendula* caused a slight increase in the relative abundance of *Ascomycota* and a slight decrease in the relative abundance of *Basidiomycota* at 4 weeks. By 7 weeks, planted soils showed slightly less *Ascomycota* and slightly more *Basidiomycota*. Interestingly, planted soils at week 4 had an approximate *Ascomycota* to *Basidiomycota* ratio of 30:1. By week 7, the *Ascomycota* to *Basidiomycota* ratio decreased to approximately 15.4:1, closer to the seed ratio of 2.8:1. These findings are consistent with the NMDS analysis showing the soil fungal community of sterilized soil planted with *B. curtipendula* resembles the seed more closely at 7 weeks post-seeding compared to 4 weeks post-seeding (Fig 2B).

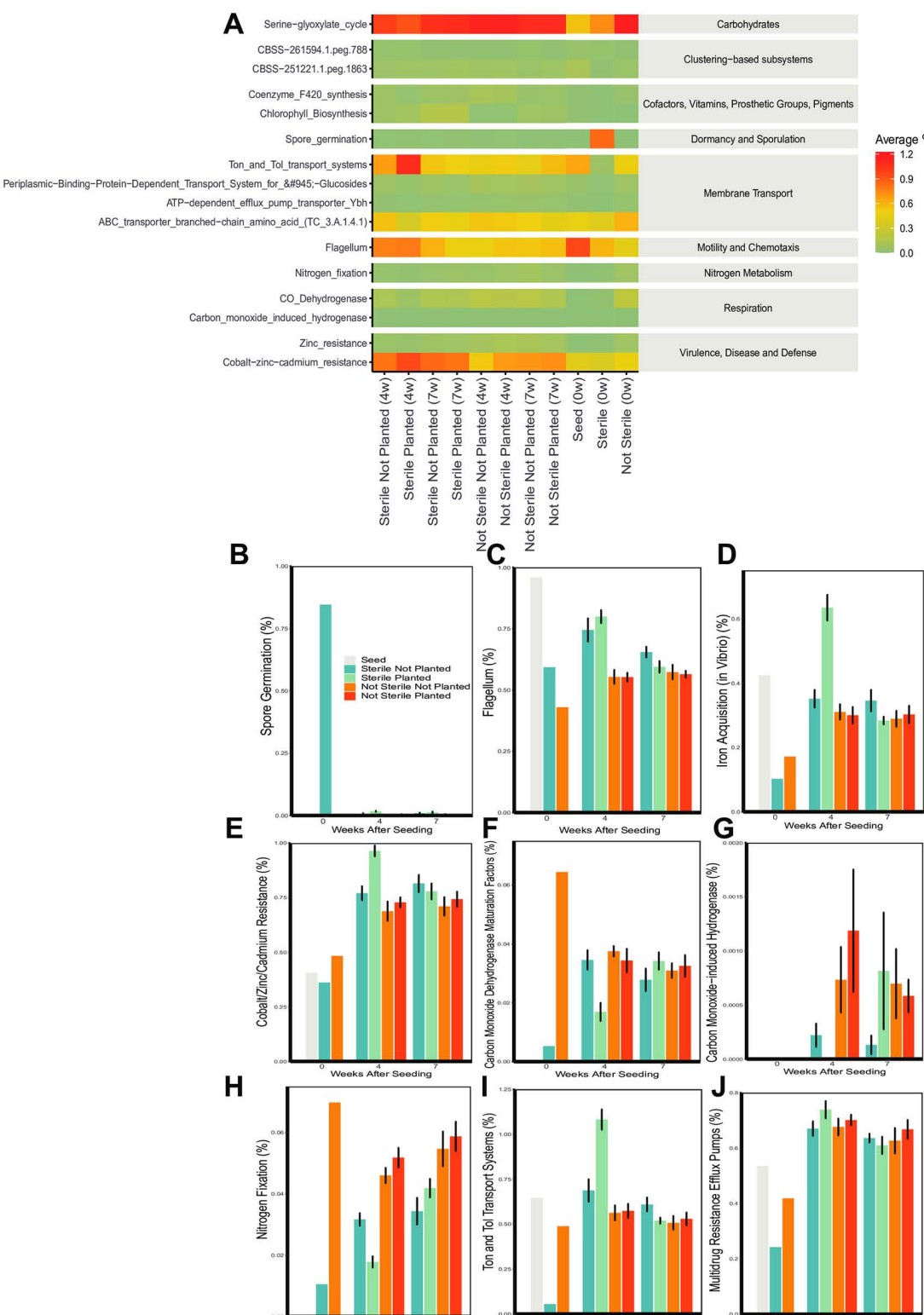

**Fig 5. Soil sterilization and planting with *Bouteloua curtipendula* changed the relative abundance of soil metabolic pathway subsystems. (A)**
Select level 3 functional subsystems for each group with the indicated level 1 metabolic system (gray background, on right). Potting soil was sterilized

via autoclaving (as described in the materials and methods, "Sterile") or left unsterilized ("Not Sterile") and then added to 72 plug trays. *B. curtipendula* was seeded at week 0 ("Planted") or left unseeded as a control ("Not Planted"). At four and 7 weeks, soil was collected for DNA purification and Illumina sequencing (as described in the materials and methods). Whole-genome sequences were mapped to level 3 functional systems in MG-RAST. **(B-J)** Mean +/- SEM of average percentage of select subsystems at weeks 0, 3, and 7 post- seeding (n = 6; week 0/seed, n = 1). The functional subsystem analyzed is indicated on the Y axis and weeks after seeding on the X axis for each of the figures.

For Bacteria, at the genus level, the sterilized soil at week 0 was almost entirely comprised of *Anoxybacillus*, *Bacillus*, *Geobacillus*, and *Paenibacillus* (Fig 8). These are all known to form heat-resistant spores and therefore not surprising to be found after sterilization. In contrast, the dominant genera in the not sterile soil at week 0 were *Mycobacterium*, *Streptomyces*, *Rhodopseudomonas*, *Salinispora*, *Mesorhizobium*, and *Bradyrhizobium* (Fig 8).

Soil sterilization impacted the relative abundance of several bacterial genera at 4 weeks post-seeding. The genus *Caulobacter*, which is often associated with plants [47], was only induced by *B. curtipendula* seeded into sterile soil (Fig 9B). In addition, the plant pathogens *Acidovorax*, *Cellvibrio*, *Pseudomonas*, and *Xanthomonas* were all overrepresented in sterilized soils planted with *B. curtipendula* (Fig 9A-D). In contrast, *Rhodopseudomonas*, which is often categorized as a plant growth promoting bacteria, was significantly reduced in the sterilized soils planted with *B. curtipendula* compared to the not sterilized planted soils (Fig 9E). However, by week 7, *Rhodopseudomonas* relative abundance greatly increased the sterilized soils planted with *B. curtipendula* and *Rhodopseudomonas* was similarly abundant across all groups. Likewise, the potential plant pathogens *Acidovorax*, *Cellvibrio*, and *Xanthomonas* were no longer elevated in the sterilized soils by week 7 (Fig 9A-D).

*Cercophora* was the predominant fungal genus in the sterilized soil at week 0 (Figs 8B, 9G), while *Mycothermus* was the predominant fungal genus in the not sterilized soil at week 0 (Figs 8B, 9I). *Epicoccum* and *Alternaria* were the most abundant fungal genera on the seed at week 0 (Figs 8B, 9F, H). *Cercophora* continued to be the dominant fungal genus in the sterilized soil at 4 weeks in both the planted and unplanted soil (Figs 8B, 9G). By 7 weeks in sterilized planted soils, a decrease in *Cercophora* relative abundance coincided with an increase in *Epicoccum, Alternaria,* and *Immersiella*. In the sterilized soils not planted with *B. curtipendula*, *Cercophora* continued to be highly abundant at 7 weeks, with no apparent increase in *Epicoccum*, *Alternaria*, or *Immersiella*. Interestingly, *Cercophora*, *Immersiella*, and *Epicoccum* remained low in abundance in the not sterile soils throughout the experiment (Figs 8B, 9G, H). Instead, *Mycothermus* remained the predominant fungal genus in the not sterile soil through week 7 (Figs 8B, 9I). However, while *Mycothermus* decreased in relative abundance over time in the not planted soil, the genus increased in relative abundance from 4 weeks to 7 weeks in planted soil (Fig 9I). The observation that seed-associated fungi (*Alternaria* and *Epicoccum*) are good at colonizing the sterile soil, but not the not sterile soil is consistent with the earlier NMDS overall community observations. This could indicate some form of competitive exclusion of *Alternaria* and *Epicoccum* in the not-sterile soil by species that were already present from the start however further detailed studies on comparative growth rates of these species will be needed to clarify this.

## Discussion

In this study, we show how transient perturbation of the soil microbial community through autoclave sterilization impairs the growth of the prairie grass *B. curtipendula. B. curtipendula* seeded into sterilized soil had a slower germination, fewer grass blades and shorter blade length throughout the course of a 7-week controlled experiment, but the difference was most notable in the first 4 weeks post-seeding. Although sterilization was transient, the soil microbial community did not immediately return to a pre- sterilization state. Instead, planting *B. curtipendula* seed into the sterilized soil caused a strong bacterial community divergence at 4 weeks that eventually stabilized by 7 weeks post-seeding (Fig 2A). In contrast, the fungal community divergence in the sterilized soil planted with *B. curtipendula* was slower, not being detectable until 7 weeks post- seeding (Fig 2B). We did not test the soil for all physical and chemical factors after autoclaving, and some

## A  Whole Genome: Phylum

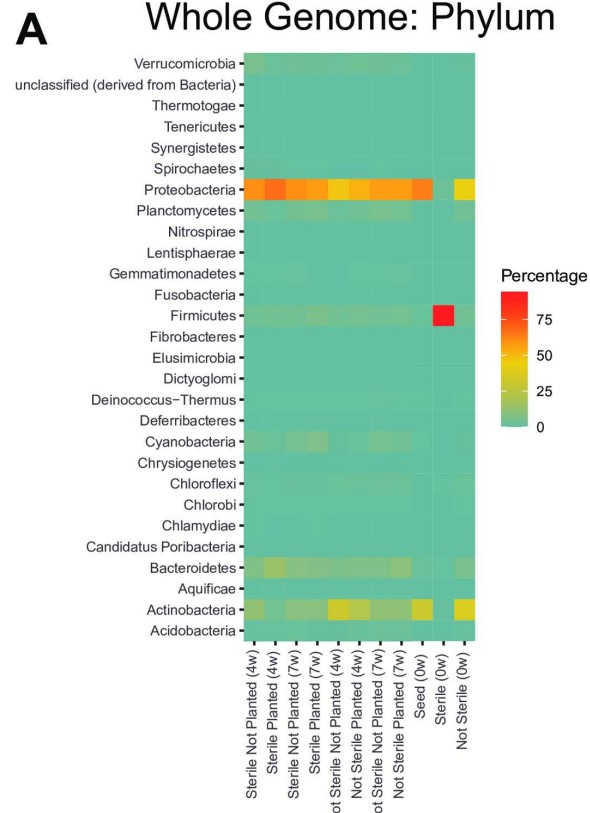

## B  Fungi Targeted

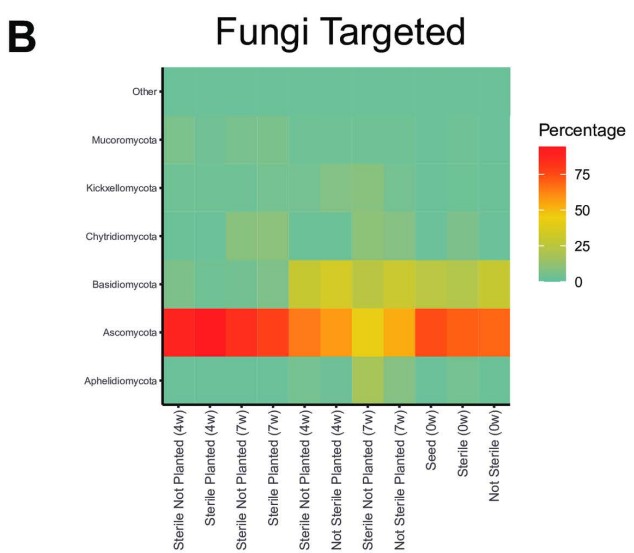

**Fig 6. Soil sterilization and planting *Bouteloua curtipendula* alters the relative abundance of soil bacterial and fungal phyla.** Potting soil was sterilized via autoclaving (as described in the materials and methods, "Sterile") or left unsterilized ("Not Sterile") and then added to 72 plug trays. *B. curtipendula* was seeded at week 0 ("Planted") or left unseeded as a control ("Not Planted"). At four and 7 weeks, soil was collected for DNA purification and Illumina sequencing. Whole-genome sequences **(A)** or fungus ITS targeted sequences **(B)** of phyla representing greater than the threshold of 2% of the total rarefied reads are represented. Phyla below the threshold were pooled into "other." Colors indicate the average percentage each phylum makes up within a group (n = 6; week 0/seed, n = 1).

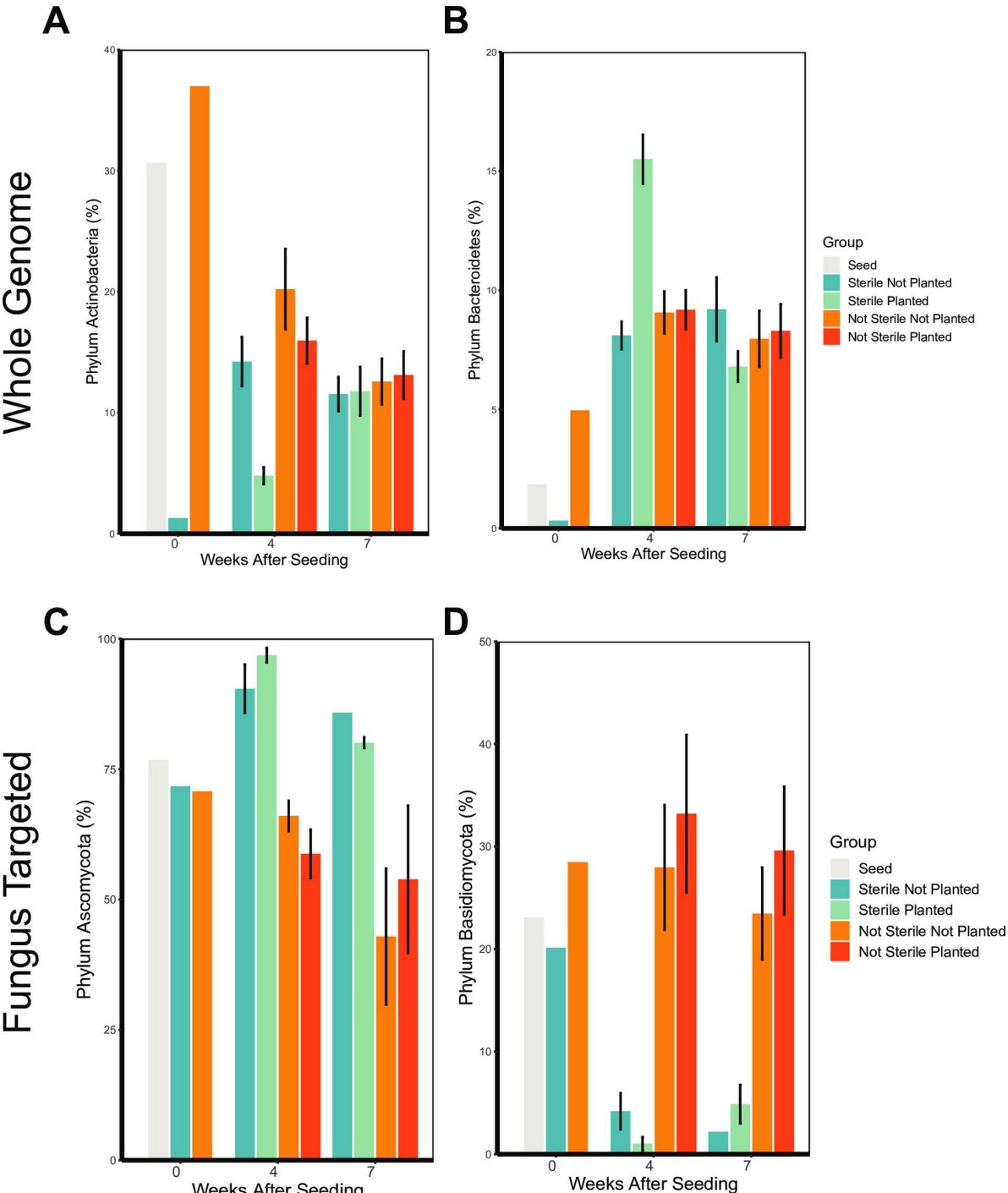

**Fig 7. Soil sterilization and planting with *Bouteloua curtipendula* changes the relative abundance of bacterial phyla *Actinobacteria* and *Bacteroidetes* and fungal phyla *Ascomycota* and *Basidiomycota*. bacterial and fungal phyla.** Potting soil was sterilized via autoclaving (as described in the materials and methods, "Sterile") or left unsterilized ("Not Sterile") and then added to 72 plug trays. *B. curtipendula* was seeded at week 0 ("Planted") or left unseeded as a control ("Not Planted"). At four and 7 weeks, soil was collected for DNA purification and Illumina sequencing. Bars represent mean +/- SEM of bacterial phyla **(A-B)** *Actinobacteria* **(A)** and *Bacteroidetes* **(B)** or fungal phyla **(C-D)** *Ascomycota* **(C)** and (*Basidiomycota*) **(D)** (n = 6; week 0/seed, n = 1).

## A

### Whole Genome: Genus

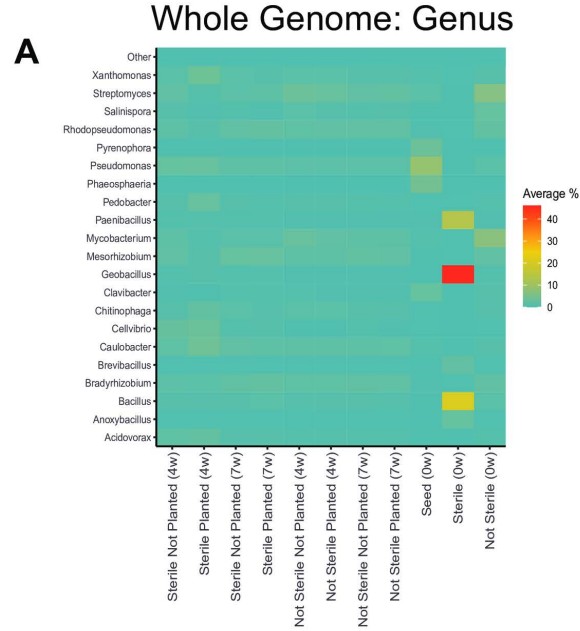

## B

### Fungus Targeted: Genus

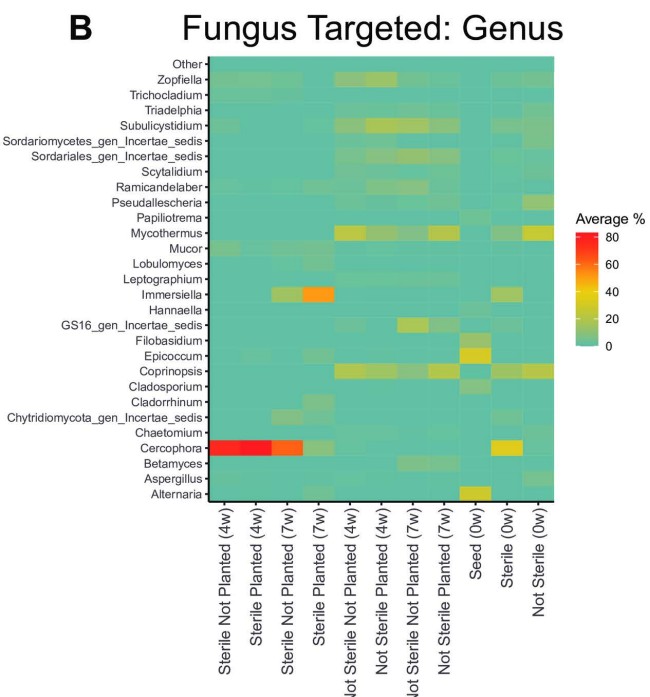

**Fig 8. Soil sterilization and planting with *Bouteloua curtipendula* alters the relative abundance of soil bacterial and fungal genera.** Whole-genome sequences **(A)** or fungus ITS targeted sequences **(B)** of genera representing greater than the threshold of 2% of the total rarefied reads are represented. Phyla below the threshold were pooled into "other." Colors indicate the average percentage each genus makes up within a group (n = 6; week 0/seed, n = 1). Potting soil was sterilized via autoclaving (as described in the materials and methods, "Sterile") or left unsterilized ("Not Sterile") and then added to 72 plug trays. *B. curtipendula* was seeded at week 0 ("Planted") or left unseeded as a control ("Not Planted") and soil samples were collected at 4 and 7 weeks, as indicated on the X axis.

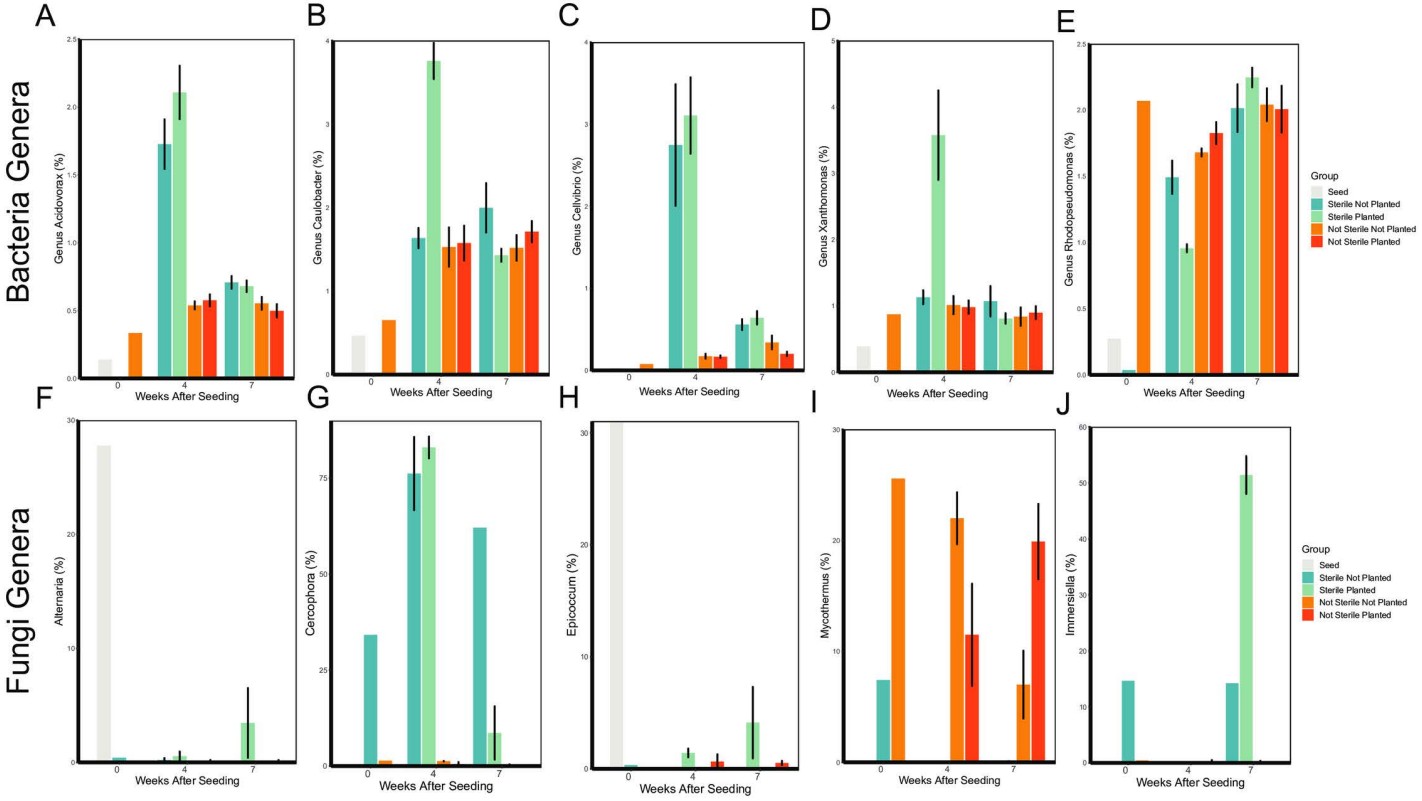

**Fig 9. Soil sterilization and planting with *Bouteloua curtipendula* alters the relative abundance of specific bacterial and fungal genera.** Bacterial (**A-E**) or Fungal (**F-J**) genera that were analyzed are indicated on the Y axis and sampling time is indicated on the X axis. Potting soil was sterilized via autoclaving (as described in the materials and methods, "Sterile") or left unsterilized ("Not Sterile") and then added to 72 plug trays. *B. curtipendula* was seeded at week 0 ("Planted") or left unseeded as a control ("Not Planted"). At four and 7 weeks, soil was collected for DNA purification and Illumina sequencing. Bars represent mean +/- SEM of bacterial (A-E) or fungal (F-J) genera (n = 6; week 0/seed, n = 1).

of these can possibly impact the initial growth of both microbes and plants, which is an inherent methodological constraint of sterilization studies. However, the plant growth rates are clearly restored after microbes reestablish. If the sterilization changes additional properties of the soil, the plant growth inhibition of this change is ameliorated by the recolonization of bacteria/fungi.

Although two rounds of autoclaving did not completely sterilize the soil, it did greatly reduce bacterial richness and diversity (Fig 3A, C). The remaining bacterial taxa identified through high throughput sequencing were spore forming bacilli (Fig 8A). Also, the dominant functional subsystem in sterilized soil at time 0 was spore germination (Fig 4). This indicates that the majority of bacterial species that is present in the initial days to weeks after sterilization are spore-forming taxa that survived sterilization or species with fast replication rates, such as *Anoxybacillus*, *Bacillus*, *Geobacillus*, and *Paenibacillus* (Fig 8). The genetic data are consistent with the morphology of viable bacteria that grew on nutrient agar plates following our sterilization protocols (data not shown). Fungal richness and diversity were only moderately reduced immediately following soil sterilization (Fig 3B, D). However, the low diversity and richness of fungi genera at 4 weeks post-seeding suggests the sequences detected immediately after sterilization arose from non-viable fungi. Indeed, we did not observe significant fungal growth on nutrient agar plates of sterilized soil even after one week of culturing at 20 degrees C (data not shown). Thus, the sterilization caused a significant disruption to the soil microbial communities, where the bacteria show a more acute stress response and faster recovery than fungi in the plant soil microbiome.

Our data also implies that fungal communities rely more heavily on external inoculation or plant root-associated processes for recovery. This can be seen by the increased fungal recovery in planted conditions and the fact that in planted condition in week 4 and 7, several of the fungal species (e.g., *Epicoccum* and *Alternaria*) are also found to be present on the seeds themselves.

Our results demonstrate soil microbial communities are highly dynamic and influenced by *B. curtipendula* planting following acute destabilization. While the overall bacterial community in sterilized soil was distinct from that of not sterilized soil at 4 weeks (Fig 2A), this difference was exacerbated when sterilized soils were planted by *B. curtipendula*. Sterilization of the soil eliminated most of the *Actinobacteria* (Fig 7A) and *Bacteriodetes* (Fig 7B) however their recovery was impacted by the planting of *B. curtipendula*. In sterilized soils, *B. curtipendula* significantly decreased the relative abundance of the phylum *Actinobacteria* and increased the relative abundance of the phylum *Bacteroidetes* at 4 weeks post-seeding (Fig 7A and B). In not-sterilized soils, the bacterial community was stable and not significantly impacted by seeding *B. curtipendula*. It therefore appears that planting with *B. curtipendula* has an initial inhibitory effect on the growth of *Actinobacteria*, but allows for an increased growth of *Bacteriodetes* and *Proteobacteria*. The mechanism underlying this will need further growth studies with specific bacterial phyla. However, by week 7 the relative abundance of *Actinobacteria, Bacteroidetes* and *Proteobacteria* were similar between all groups, indicating that these differences were limited to the temporal changes due to the destabilization and the planting.

We also report how *B. curtipendula* showed delayed signs of germination (Fig 1A) and grew more slowly (Fig 1B) in soils with an acutely disrupted microbiome compared to unperturbed soils (Fig 1). It is interesting to note how soil sterilization increased the relative abundance of potential plant pathogen bacteria, such as *Acidovorax* [48–50], *Cellvibrio* [51,52], and *Xanthomonas* [53,54] at 4 weeks post- seeding. The latter observation is consistent with reports of *Xanthomonas* increase in phytopathogenic biofilm formation in plant roots [55,56]. By 7 weeks post-seeding, the potentially pathogenic bacteria stabilized and returned to levels comparable to that of the not sterilized soil seeded with *B. curtipendula*. These data are consistent with our observation of impaired *B. curtipendula* growth in sterilized soil, especially during the first 4 weeks post-seeding.

Acute disruption of gut microbiota with antibiotics is known to increase the relative abundance of human pathogens [36]. Our findings suggest acute disruption of the soil microbiota also promotes plant pathogens that impede growth of plants, such as the prairie grass *B. curtipendula*. It is interesting that the bacterial communities restabilized by 7 weeks post-seeding and *B. curtipendula* growth in sterilized soil was less deficient at these later time points. It is possible that soil texture, moisture, pH, and organic mineral content (which should be largely similar between sterilized and not sterilized soil) is the major determinant of long-term soil microbial community, and acute disturbance will always be a transient disruption before returning to the long-term stable community. In other microbiome systems, such as human gut or skin microbiomes, it is known that symbiotic bacteria have the ability to inhibit pathogen colonization through several mechanisms [36,57]. A diverse soil microbiome seems to have the same protective characteristics in our present study and similar mechanisms could involve direct killing, competition for limited nutrients and the enhancement of plant protective responses. Any factor that causes a dysbiosis in the diverse microbiome (such as autoclaving or chemical or nutrient changes) allows for opportunities of pathogens to increase.

It is also possible that the fungal species aided in stabilization of the bacterial community and the reduction of potentially pathogenic bacterial taxa by 7 weeks. Indeed, we find the *B. curtipendula*-induced fungal community took 7 weeks to establish in sterilized soil (Fig 2B). At the phylum level, planting sterilized soil with *B. curtipendula* caused a slight increase in the relative abundance of *Ascomycota* and a slight decrease in the relative abundance of *Basidiomycota* which is visible at 4 weeks (Fig 7C and D). However by 7 weeks, planted soils showed less *Ascomycota* and more *Basidiomycota*, which is more similar to the seed fungal ratio. This indicates that the soil fungal community of sterilized soil planted with *B. curtipendula* resembles the seed more closely at 7 weeks post-seeding compared to 4 weeks post-seeding, which is consistent with the NMDS analysis (Fig 2B). Interestingly, one of the fungal genera that increased in *B. curtipendula*-planted

sterilized soil by 7 weeks includes *Epicoccum*, which was also highly abundant on the *B. curtipendula* seed. Thus, seed-associated fungi could be responsible for reducing the plant pathogens in the soil and thus promoting *B. curtipendula* growth. Indeed, *Epicoccum sp.* associated with other plants have been shown to produce antimicrobial compounds against phytopathogens [58]. In contrast, the other seed-associated fungi induced by 7 weeks, *Alternaria*, is most commonly associated as a fungal plant pathogen itself [59,60]. Future studies will be necessary to determine whether seed-associated fungi promote *B. curtipendula* growth through antimicrobial activity against plant pathogens in the soil.

Nevertheless, our results show the soil microbial dynamics associated with inhibition of prairie grass growth following acute soil sterilization. *B. curtipendula* seeded into sterilized soil induced more of the potentially pathogenic bacterial genera *Acidovorax*, *Cellvibrio*, and *Xanthomonas* and less of the plant growth promoting *Rhodopseudomonas* at 4 weeks post-seeding (Fig 9A-E). This overabundance of bacterial plant pathogens subsided at 7 weeks post-seeding into sterilized soil, giving way to *B. curtipendula*-induced fungal species such as *Alternaria*, *Epiccocum*, and *Immersiella* (Fig 9F,H, I).

Our study is consistent with previous studies showing interactions between plants and the soil microbial community. Much of this work involves a group of bacteria categorized as plant growth promoting bacteria (PGPB) and fungi (PGPF) [61]. Microbes known to be PGPB and PGPF can release secondary metabolites that suppress plant pathogenic bacteria and fungi in the soil, thus promoting plant growth [62]. Such is the case for the fungus *Mycothermus*, which showed a strong positive correlation with Lisianthus (*Eustoma sp.*) growth due to suppression of plant disease [63]. PGPB can also promote nutrient uptake and increase fertilizer efficiency [64]. One specific bacteria genus associated with plant growth is *Rhodopseudomonas*. When inoculated into sterilized soil, *Rhodopseudomonas* enhanced germination of tomato plants [65]. *Rhodopseudomonas* incoculation also promoted growth of Stevia [66], Brassica plants [67,68], beans (*Vigna mungo*) [69], rice [70,71], and provide systemic resistance to TMV in tobacco plants [72]. We found significantly less *Rhodopseudomonas* in sterilized soil planted with *B. curtipendula* compared to not sterilized soil planted with *B. curtipendula* at four weeks post seeding (Fig 9). In addition, we found sterilization essentially eliminated *Mycothermus* from the soils throughout the course of our experiment (Fig 9). The reduction of *Rhodopseudomonas* and *Mycothermus* in sterilized soil is associated with increased pathogenic bacterial and fungal taxa (Fig 9) and impaired *B. curtipendula* growth (Fig 1). Thus, our results suggest that a diverse and unperturbed microbial community could support prairie grass growth through multiple mechanisms: directly promoting plant growth and reducing the abundance of potential plant pathogens.

It has been shown that the microbial interaction with plant roots can result in improved nutrient and mineral uptake, help in plant-growth promotion as well a suppression of phytopathogens [73,74]. The role of plant growth promotion involves phytohormone production, nitrogen fixation, siderophore production, solubilization of inorganic substances (P, K, Zn etc.), and the microbial remediation of heavy metal toxicity (such as high cadmium in fertilizer) [74,75]. The mechanisms of suppression of phytopathogens can involve direct methods of production of antibacterial or antifungal metabolites or production of wall-degrading enzymes such as chitinase, but also involves indirect routes such as iron chelation and depletion from the rhizosphere [76]. A search of the known genomes of *Rhodopseudomans* strains [77] shows that several strains contain genes for 1-aminocyclopropane-1-carboxylate (ACC) deaminase, which is an enzyme known to reduces plant stress hormones [28], and genes for enzymes involved in indole-3-acetic acid (IAA) and 5-aminolevulinic acid (ALA) production, which may also be one of the mechanisms of plant growth enhancement [65]. Certain strains of *Rhodopseudomonas* have been shown to have plant growth promoting effects by stimulating nitrogen uptake and increasing IAA levels [68], which is consistent with our observations here. Unfortunately the current resolution of metagenomic sequencing is unable to confidently identify to the species level, nevertheless it is likely that the *Rhodopseudomonas* strains involved here have one or several of the characteristics above that contribute to the plant growth promoting effect.

The subsystem 'Zinc resistance' is one of the Virulence, Disease, and Defense (Subsystem 1) functions, that was elevated in sterile planted soil at 4 weeks compared to other groups (Fig 5). The specific enzymes related to this (Function level) were the sensor protein and response regulator of the Sigma-54 two component system. The Sigma factor 54 is a

central regulator in many pathogenic bacteria and has been linked to important functional traits such as motility, virulence, host colonization, and biofilm formation [78]. The higher level of this function in the sterile soil is consistent with our observation of higher levels of potential pathogens in the sterile soil, before a more diverse microbiome becomes established.

Overall, the majority of metabolic pathways were similar between groups and conditions, aside from the sterilized soil at week 0, which indicates an overall metabolic stability in the microbiome, where the major soil biochemical reactions can be performed by a diverse group of bacteria. The presence of similar metabolic pathway profiles across soil conditions suggests the possibility for functional redundancy in the microbial community. It is therefore expected that different groups of taxa can maintain core ecosystem services, and for example when studying the growth of *Bouteloua* in native prairie environments we may observe different microbiome compositions than in a controlled greenhouse setting. However, the core supporting functional metabolic pathways will likely be very similar. It is therefore imperative to view these microbial communities as networks of genes, proteins and metabolic signals, rather than a group of very specific species. Nevertheless, knowing the presence and understanding the function of beneficial species described in this study and other ones, provides not only a broader understanding of the diversity, but can also contribute to improving soil health and soil microbiome resilience to acute and long-term disturbances.

Ultimately, our results suggest that functional and diverse microbial soil communities could aid in the establishment of prairie grasses and subsequently increase the success of urban prairie plantings. Native plant species promote broad ecological functions in habitat areas and these functions are reduced by invasive species. However, microbial communities associated with invasive species can favor invasive grasses and impede native grasses [79]. During prairie restoration projects, or when converting urban settings into native gardens, there will be acute and long-term disturbances in the soil microbiome composition. These involve the initial removal or killing of invasive species or turf grass by either chemical or physical methods, or the addition of new soil to the area. These conditions are different for various projects, however not unlike our sterilization experiment, and will cause acute disturbances in the soil microbiome. Our study shows that the time of the transition period and recolonization after an acute disturbance in an environmental setting impacts the establishment of the plants and can take up to 7 weeks for a microbial restoration. This stresses the importance of establishing a resilient microbial ecosystem as early as possible during ecological restoration projects. This will help to avoid the growth of potential plant pathogens that could delay or impair the growth of the native plants. Another study found that soil sterilization increased the success of invasive *Bromus tectorum* infiltrating into *Bouteloua gracilis* (a plant closely related to *B. curtipendula* used in our study) [37]. These findings, combined with the results of our study, emphasize the importance of soil microbial communities in promoting highly functional native plant communities. Furthermore, our study indicates that the bacteria are involved in the initial establishment of beneficial conditions which pave the way for a solid fungal and plant seedling development. Understanding these microbiome-plant relationships in native *B. curtipentula* opens up possibilities for incorporating target strains that help the colonization and growth of this native grass in restored native habitats. Soil health is heavily researched and recognized as an essential component for food crop production, and more holistic approaches that take into account microbial optimization have been proposed for sustainable agriculture [80,81]. Soil microbial health is similarly important for native plant establishment in urban or prairie settings, and microbial optimization in the case of *Bouteloua* or other native plants could help their long-term establishment. One way to approach this might be to incorporate these growth-promoting bacterial and fungal strains into seeds, similar to the ENDOSEED concept that has been performed in crop plants [27,82]. To our knowledge such concepts have not been performed with native grasses like *Bouteloua*, and the long- term effects on plant roots and exposure to varied environmental conditions would certainly have to be tested in future experiments. However our initial study indicates that similar approaches could provide benefits to the initial establishment of the native grass during restoration projects. Irrespective of the application method, efforts to promote native habitat spaces should consider concepts which include native-plant promoting soil microbial communities to ensure the greatest habitat functionality.

## Supporting information

**S1 File. Supplemental BoutelouaMicrobiome MG-RAST-script-16-July-2025.**
(TXT)

## Acknowledgments

We would like to thank Julian Ramirez for his assistance in maintaining the greenhouse and plants used in this experiment.

## Author contributions

**Conceptualization:** Daisy Ochoa-Rojas, John A. Kyndt, Tyler C. Moore.

**Data curation:** Alisiara Hobbs, Christine E. Humphrey, Tyler C. Moore.

**Formal analysis:** Alisiara Hobbs, Daisy Ochoa-Rojas, Christine E. Humphrey, John A. Kyndt, Tyler C. Moore.

**Funding acquisition:** John A. Kyndt, Tyler C. Moore.

**Investigation:** Alisiara Hobbs, Daisy Ochoa-Rojas, Christine E. Humphrey, John A. Kyndt, Tyler C. Moore.

**Methodology:** Alisiara Hobbs, Christine E. Humphrey, John A. Kyndt.

**Project administration:** Tyler C. Moore.

**Resources:** Daisy Ochoa-Rojas, John A. Kyndt.

**Software:** Alisiara Hobbs, Christine E. Humphrey, John A. Kyndt, Tyler C. Moore.

**Supervision:** John A. Kyndt, Tyler C. Moore.

**Validation:** Alisiara Hobbs, Daisy Ochoa-Rojas, Christine E. Humphrey, Tyler C. Moore.

**Visualization:** Tyler C. Moore.

**Writing – original draft:** John A. Kyndt, Tyler C. Moore.

**Writing – review & editing:** Alisiara Hobbs, Daisy Ochoa-Rojas, Christine E. Humphrey, John A. Kyndt, Tyler C. Moore.

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
