## [Decision Letter · Decision Letter 0]

19 Feb 2025

Dear Dr. Kyndt,

We look forward to receiving your revised manuscript.

Kind regards,

Rajesh Singh Rathore

Academic Editor

PLOS ONE

Journal Requirements:

2. In the online submission form, you indicated that your data is available only on request from a third party. Please note that your Data Availability Statement is currently missing the contact details for the third party, such as an email address or a link to where data requests can be made. Please update your statement with the missing information.

Additional Editor Comments:

Dear Dr. Kyndt,

I hope this message finds you well.

I am writing to inform you that after careful consideration of the reviewer comments, we would like to invite you to revise and resubmit your manuscript entitled "Soil Microbiome Perturbation Impedes Growth of Bouteloua curtipendula and Increases Relative Abundance of Soil Microbial Pathogens" for major revision. Both Reviewer 1 and Reviewer 2 have provided constructive feedback that, if addressed, will significantly improve the quality and clarity of your manuscript.

Major Points for Revision:

Abiotic Factors and Sterilization Effects

Microbial Recolonization and Ecological Relevance

Sequencing and Data Quality

Statistical Analysis and Experimental Design

Microbial Roles and Plant Growth

Sequencing Issues and Replicates etc.

Sincerely

-Rajesh

Reviewers' comments:

Reviewer's Responses to Questions

**Comments to the Author**

1. Is the manuscript technically sound, and do the data support the conclusions?

Reviewer #1: Yes

Reviewer #2: Partly

2. Has the statistical analysis been performed appropriately and rigorously?

Reviewer #1: Yes

Reviewer #2: N/A

3. Have the authors made all data underlying the findings in their manuscript fully available?

Reviewer #1: Yes

Reviewer #2: Yes

4. Is the manuscript presented in an intelligible fashion and written in standard English?

Reviewer #1: Yes

Reviewer #2: Yes

Reviewer #1: The manuscript titled here “Soil microbiome perturbation impedes growth of Bouteloua curtipendula and increases relative abundance of soil microbial pathogens”. The MS provides significant insights into the impact of acute soil microbial community perturbations, such as autoclave sterilization, on the growth dynamics of Bouteloua curtipendula and subsequent microbial succession. The findings highlight critical interactions between soil bacteria, fungi, and plant growth, contributing to the broader understanding of microbial ecology and plant-microbe interactions in restored native habitats. The integration of genomic analysis with plant growth metrics strengthens the study's conclusions.

However, authors should address the following comments:

• Please avoid initiating the Introduction section with “ Although”. Consider beginning with a strong, context-setting statement that highlights the relevance of soil microbial perturbations in ecological studies or restoration efforts.

• MM section- “soil samples were diluted in distilled water” – please mention how much soil was taken in how much water.

• “1 g of soil on LB nutrient agar plates”- I g of soil was directly plated or suspended in some solution before plating. Please clarify.

• The manuscript states that whole-genome sequencing was used for bacterial identification, whereas fungal identification relied on ITS region sequencing. Clarify why 16S rRNA gene sequencing was not employed for bacterial community analysis to maintain consistency in approaches. Discuss the rationale for choosing these methods.

• Some sentences are long and can be broken into shorter for e.g. "In the non-sterilized samples, soil planted with B. curtipendula had a similar overall bacterial community to non-planted soil..." can be split to improve readability. Please check throughout the MS.

• Ensure consistent use of terms like "non-sterilized" vs. "not sterilized." Pick one term and stick with it throughout.

• Similarly, ensure terms like "planted soil" vs. "planted condition" are uniformly used

• The distinct recovery timelines for bacterial (4 weeks) and fungal (7 weeks) communities are noteworthy. However, the study does not delve deeply into potential ecological mechanisms underpinning these differences. Could the bacterial recovery be attributed to faster replication rates or the presence of spore-forming taxa that survived sterilization? Conversely, do fungal communities rely more heavily on external inoculation or plant root-associated processes for recovery?

• The observed transient increase in potential pathogenic bacteria (e.g., Acidovorax, Xanthomonas) following sterilization and their subsequent decline suggest competitive exclusion or suppression by other microbial taxa. The authors should expand on the mechanisms driving pathogen suppression.

• Were any functional genes associated with PGPB activity (e.g., nitrogen fixation or siderophore production) detected in metagenomic analyses, even at low abundance?

• The observation of similar metabolic pathway profiles across soil conditions, aside from sterile soils at time 0, suggests functional redundancy in the microbial community. Could this indicate that even after disruption, soil microbiomes can maintain core ecosystem services, albeit with altered taxa? Discussing this in the context of resilience and ecosystem functionality could strengthen the manuscript

• In some of the figures, the error bar is out of the scale. Please correct.

Reviewer #2: Comments to the authors:

How did the study ensure that the observed differences in B. curtipendula growth were solely due to soil microbial communities rather than other abiotic factors altered by sterilization (e.g., nutrient availability, soil structure, or chemical changes)?

The study found that bacterial communities in sterilized soil eventually became similar to those in non-sterilized soil. What mechanisms or sources contributed to this microbial recolonization, and could this transition period impact long-term plant establishment and survival in prairie restorations?

Given that urban grasslands and prairie restorations are the intended application areas, how do the study's findings translate to real-world settings with naturally occurring microbial heterogeneity and environmental stressors (e.g., drought, pollution, competition from invasive species)?

Is the method of microbial perturbation (autoclave sterilization) a reliable proxy for real-world soil disturbances in urban environments?

Are the high-throughput sequencing methods used in the study robust enough to capture the functional roles of microbial communities in pocket prairie soil?

How was it ensured that the sterilization process did not alter the soil’s physical and chemical properties beyond microbial depletion, potentially affecting plant growth independently of microbial presence?

How were plant height and blade numbers recorded to minimize measurement error and observer bias? Were multiple measurements taken per plant, and was there any calibration or standardization of the measurement tools?

Was any quality assessment (e.g., Bioanalyzer, Qubit) performed before sequencing to ensure high-quality input DNA for WGS and ITS sequencing?

Can the authors provide more details on the distribution of sequencing depth across different samples? Were there significant variations, and if so, how were they accounted for in downstream analyses?

Was any rarefaction analysis performed to ensure sufficient sequencing depth for comparative analyses?

How does the accuracy of MG-RAST compare to other taxonomic classification tools such as Kraken2, Kaiju, or MetaPhlAn for metagenomic analysis?

Were any de novo assembly-based methods used alongside MG-RAST to improve species-level classification?

What were the specific reasons for the failed sequencing of the replica sample (sterile soil with B. curtipendula)? Could this indicate issues with DNA extraction or library preparation?

Were biological replicates used in addition to technical replicates to assess the reproducibility of taxonomic assignments?

The study reports significant differences in the germination rates of B. curtipendula in sterilized versus non-sterilized soils. Could other abiotic factors, such as changes in nutrient availability or soil structure post-autoclaving, have influenced germination in addition to microbial community alterations?

The bacterial community in sterilized soils eventually converges towards that of non-sterilized soil by week 7. Was any quantitative measure (e.g., alpha or beta diversity indices) used to evaluate the pace of microbial succession?

The study identifies shifts in metabolic pathways (e.g., carbohydrate metabolism, dormancy, and sporulation) following soil sterilization. Can these shifts be directly linked to changes in plant growth parameters?

The study uses one-way ANOVA for comparing growth parameters. Were assumptions of normality and homogeneity of variance tested before applying ANOVA?

The findings suggest that seed-associated fungi (Alternaria, Epicoccum) preferentially colonized sterile soil but not non-sterile soil. Could this be due to competitive exclusion in non-sterile soil, and was any additional sequencing of seed-associated microbes conducted to further explore this hypothesis?

How did the researchers confirm that the initial bacterial and fungal communities were representative of typical prairie soil conditions? Could there be a potential bias introduced by the sterilization process that alters the baseline microbial community differently than anticipated?

How were the control groups structured, particularly regarding soil conditions? Were there any unsterilized soil groups where B. curtipendula was not seeded to isolate the effects of soil sterilization alone versus the interaction with the grass?

The study suggests that Rhodopseudomonas and Mycothermus play roles in promoting plant growth. How were these roles verified experimentally, beyond correlation? Is there direct evidence linking the presence or absence of these microbes to the observed changes in B. curtipendula growth?

What mechanisms are proposed for the transient increase in potentially pathogenic bacterial genera like Acidovorax, Cellvibrio, and Xanthomonas in sterilized soils? Could other environmental factors, such as soil nutrient changes due to autoclaving, contribute to this shift?

How did the authors ensure the accuracy of the metagenomic sequencing data, particularly given the challenges of identifying microbes at the species level? Were there any potential biases in the sequencing or data analysis that might affect the results?

The authors suggest that incorporating growth-promoting microbes into seeds could enhance the establishment of native grasses. How would this approach account for the complex and dynamic interactions between soil microbes and plant roots over time, especially in varied environmental conditions?

**Do you want your identity to be public for this peer review?** For information about this choice, including consent withdrawal, please see our Privacy Policy

Reviewer #1: No

Reviewer #2: No

---

## [Author Response · Author response to Decision Letter 1]

2 Apr 2025

These responses are provided as a separately uploaded document, but are also copied here as requested:

PONE-D-24-43622

Soil microbiome perturbation impedes growth of Bouteloua curtipendula and increases relative abundance of soil microbial pathogens

Dear Editor,

We appreciate the reviewers’ comments and have provided our rebuttal to each of the points below. In addition we provided a marked-up document that highlights each of the changes made.

Reviewer #1: The manuscript titled here “Soil microbiome perturbation impedes growth of Bouteloua curtipendula and increases relative abundance of soil microbial pathogens”. The MS provides significant insights into the impact of acute soil microbial community perturbations, such as autoclave sterilization, on the growth dynamics of Bouteloua curtipendula and subsequent microbial succession. The findings highlight critical interactions between soil bacteria, fungi, and plant growth, contributing to the broader understanding of microbial ecology and plant-microbe interactions in restored native habitats. The integration of genomic analysis with plant growth metrics strengthens the study's conclusions.

However, authors should address the following comments:

• Please avoid initiating the Introduction section with “ Although”. Consider beginning with a strong, context-setting statement that highlights the relevance of soil microbial perturbations in ecological studies or restoration efforts.

This has been updated

• MM section- “soil samples were diluted in distilled water” – please mention how much soil was taken in how much water.

500 mg of soil was diluted in 5 mL of water. This was mentioned elsewhere in the manuscript, but is now moved to the correct section.

• “1 g of soil on LB nutrient agar plates”- I g of soil was directly plated or suspended in some solution before plating. Please clarify.

The 1g of soil was plated directly onto the agar. This is now clarified in the MM section.

• The manuscript states that whole-genome sequencing was used for bacterial identification, whereas fungal identification relied on ITS region sequencing. Clarify why 16S rRNA gene sequencing was not employed for bacterial community analysis to maintain consistency in approaches. Discuss the rationale for choosing these methods.

16S rRNA sequencing focuses on identifying and quantifying bacterial species and community composition but does typically not provide information down to the strain level nor does it give metabolic insights, while whole genome sequencing (WGS) provides a more comprehensive view of the microbiome, including functional genes and potential for strain-level resolution and antimicrobial resistance detection. However when using a WGS approach, the fungal species are often underrepresented (due to the abundance of bacteria) and therefore a more targeted approach is beneficial. This reasoning is now added to the manuscript.

• Some sentences are long and can be broken into shorter for e.g. "In the non-sterilized samples, soil planted with B. curtipendula had a similar overall bacterial community to non-planted soil..." can be split to improve readability. Please check throughout the MS.

Several sentences have been restructured and split for clarity throughout the MS.

• Ensure consistent use of terms like "non-sterilized" vs. "not sterilized." Pick one term and stick with it throughout.

The term not-sterilized is now use throughout the MS consistently (including in the figures)

• Similarly, ensure terms like "planted soil" vs. "planted condition" are uniformly used

This is now updated and ‘Planted soil’ is used consistently.

• The distinct recovery timelines for bacterial (4 weeks) and fungal (7 weeks) communities are noteworthy. However, the study does not delve deeply into potential ecological mechanisms underpinning these differences. Could the bacterial recovery be attributed to faster replication rates or the presence of spore-forming taxa that survived sterilization? Conversely, do fungal communities rely more heavily on external inoculation or plant root-associated processes for recovery?

Yes, we had eluted to both of these points in the results and the discussions, but we now have expanded on these more in the discussion section.

• The observed transient increase in potential pathogenic bacteria (e.g., Acidovorax, Xanthomonas) following sterilization and their subsequent decline suggest competitive exclusion or suppression by other microbial taxa. The authors should expand on the mechanisms driving pathogen suppression.

We had mentioned earlier in the discussion that fungal growth (for example Epicoccum sp.) has been associated with production of antimicrobial compounds against phytopathogens. We have now also expanded on the importance of the bacterial diverse microbiome to create an environment that is less conducive for growth of pathogenic organisms in the discussion.

• Were any functional genes associated with PGPB activity (e.g., nitrogen fixation or siderophore production) detected in metagenomic analyses, even at low abundance?

Those genes are certainly present in the analysis, but the gene abundances are not significantly different, and since we didn’t measure actual expression levels we could not expand on any possible production level differences.

• The observation of similar metabolic pathway profiles across soil conditions, aside from sterile soils at time 0, suggests functional redundancy in the microbial community. Could this indicate that even after disruption, soil microbiomes can maintain core ecosystem services, albeit with altered taxa? Discussing this in the context of resilience and ecosystem functionality could strengthen the manuscript

We have expanded on this in the discussion.

• In some of the figures, the error bar is out of the scale. Please correct.

In some figures the error bars are approaching or reaching the outside frame. Therefore we removed the outside frame from the graphs in Figures 3 A-D. That shows the entire graph and error bars now.

Reviewer #2: Comments to the authors:

How did the study ensure that the observed differences in B. curtipendula growth were solely due to soil microbial communities rather than other abiotic factors altered by sterilization (e.g., nutrient availability, soil structure, or chemical changes)?

We did not test for all these factors and some of these can possibly impact the initial growth (of both microbes and plants). But if so, the plant growth rates are restored after microbes reestablish. If the sterilization changes additional properties of the soil, the plant growth inhibition of this change is ameliorated by the recolonization of bacteria/fungi. We have expanded on this briefly in the discussion section.

The study found that bacterial communities in sterilized soil eventually became similar to those in non-sterilized soil. What mechanisms or sources contributed to this microbial recolonization, and could this transition period impact long-term plant establishment and survival in prairie restorations?

In an open environmental system (even within a greenhouse setting) there is an abundance of environmental organisms that given the right conditions will propagate. Even minimal amounts of survival organisms in autoclaved soil can quickly recolonize. It is difficult to speculate in the current model to specify the source or mechanism of recolonization, which would require detailed follow up studies. The timing of the transition period and recolonization after an acute disturbance in an environmental setting would certainly impact the establishment of the plants. We have expanded upon this more in the discussion.

Given that urban grasslands and prairie restorations are the intended application areas, how do the study's findings translate to real-world settings with naturally occurring microbial heterogeneity and environmental stressors (e.g., drought, pollution, competition from invasive species)?

We have expanded upon this in the discussion and made comparisons between our simulated acute disturbance and the processes that occur during prairie restoration or urban native garden projects.

Is the method of microbial perturbation (autoclave sterilization) a reliable proxy for real-world soil disturbances in urban environments?

We have expanded upon this in the discussion and made comparisons between our simulated acute disturbance and the processes that occur during prairie restoration or urban native garden projects.

Are the high-throughput sequencing methods used in the study robust enough to capture the functional roles of microbial communities in pocket prairie soil?

The high-throughput sequencing methods used in this study are the current standard for metagenomics studies in complex microbiomes in many environments. Our studies did not address any specific functional roles of specific species or species interactions in these environments, which would require follow up transcriptomics or proteomics-based studies or follow-up growth experiments with specific bacterial or fungal species. These are great follow up studies, but currently fall beyond the scope of this paper.

How was it ensured that the sterilization process did not alter the soil’s physical and chemical properties beyond microbial depletion, potentially affecting plant growth independently of microbial presence?

This is possible. But if so, the plant growth rates are restored after microbes reestablish. If the sterilization changes additional properties of the soil, the plant growth inhibition of this change is ameliorated by the recolonization of bacteria/fungi. We have expanded on this briefly in the discussion section.

How were plant height and blade numbers recorded to minimize measurement error and observer bias? Were multiple measurements taken per plant, and was there any calibration or standardization of the measurement tools?

All measurements were performed by a single individual to ensure consistency over time. At the start of the study, measurements were compared between two individuals to conclude minimal impact on between-measurer variability. The calibration tools were standard supply rulers that did not require calibration. We have now included this in the MM section.

Was any quality assessment (e.g., Bioanalyzer, Qubit) performed before sequencing to ensure high-quality input DNA for WGS and ITS sequencing?

Yes, we performed Qubit analysis in addition to the Nanodrop measurements of the DNA and the prepared libraries before sequencing. This is now added to the Material and Methods.

Can the authors provide more details on the distribution of sequencing depth across different samples? Were there significant variations, and if so, how were they accounted for in downstream analyses?

There was some variation in the read counts per sample (ranges are provided in the paper) although all samples used in the analysis had sufficient minimal read coverage. MG-RAST uses DEseq for normalization, which we had indicated in the paper. DEseq is an R package used to analyze count data from high-throughput sequencing assays. DESeq, as it has been shown to outperform other methods of normalization, in particular those that use any sort of linear scaling.

Was any rarefaction analysis performed to ensure sufficient sequencing depth for comparative analyses?

Yes, rarefaction analysis was performed on both the MG-RAST and BaseSpace Metagenomics output, and all samples were found to be with sufficient sequencing depth for further analysis (with the exception of the one sample that generated low data as we explained in the text). We have now included a more detailed section in the Material and Methods about the rarefaction, normalization and other statistical analysis that was performed.

How does the accuracy of MG-RAST compare to other taxonomic classification tools such as Kraken2, Kaiju, or MetaPhlAn for metagenomic analysis?

MG-RAST is a well-established tool for reference-based classification and provides an online metagenomic analysis interface that includes data uploading, QC and alignment with reference databases. It has the option of a no command line use through the web-portal. The advantage of MG-RAST is that not only taxonomic classification is provided, but also functionals subsystem analysis as part of the comprehensive data analysis. This does lead to longer computing time necessary for aligning contigs to a reference. It is a highly accurate method but has the disadvantage of being slow. To speed up the process, alternative methods such as Kraken have been developed to replace the direct alignment of a query against a reference database by a fast-lookup method of fixed-length k-mers extracted from the query. Other studies have shown that Kraken 2 and MG-RAST generate comparable results and that a reliable high-level overview of sample is generated irrespective of the pipeline selected. As long as up to date databases are used for classification. Many analysis packages are now available and their accuracy and validity have been compared extensively in the last years. Here are some examples of publications that used MG-RAST in their comparisons to other bioinformatic resources:

https://pubmed.ncbi.nlm.nih.gov/32382536/

https://www.sciencedirect.com/science/article/pii/S2001037021004931

https://pmc.ncbi.nlm.nih.gov/articles/PMC5861821/

https://pmc.ncbi.nlm.nih.gov/articles/PMC7641418/#R205

Were any de novo assembly-based methods used alongside MG-RAST to improve species-level classification?

No

What were the specific reasons for the failed sequencing of the replica sample (sterile soil with B. curtipendula)? Could this indicate issues with DNA extraction or library preparation?

The DNA and library prep assessment looked fine, but resulted in low sequencing output. The sequencing of that sample was never repeated since we had 5 other valid replicas.

Were biological replicates used in addition to technical replicates to assess the reproducibility of taxonomic assignments?

All of the samples were taken from their individual growth experiment, so they are all biological replicates.

The study reports significant differences in the germination rates of B. curtipendula in sterilized versus non-sterilized soils. Could other abiotic factors, such as changes in nutrient availability or soil structure post-autoclaving, have influenced germination in addition to microbial community alterations?

As mentioned above, this is possible. But if so, the plant growth rates are restored after microbes reestablish. If the sterilization changes additional properties of the soil, the plant growth inhibition of this change is ameliorated by the recolonization of bacteria/fungi. We have expanded on this briefly in the discussion section.

The bacterial community in sterilized soils eventually converges towards that of non-sterilized soil by week 7. Was any quantitative measure (e.g., alpha or beta diversity indices) used to evaluate the pace of microbial succession?

The alpha diversity is shown in the Shannon index in Figure 3. The beta diversity is the change in community across sample conditions, which we show in Figure 2 ordination plots. Although we didn't quantitate microbial diversity changes, we did use Adonis multivariate ANOVA to statistically test the change in community composition across time points and conditions.

The study identifies shifts in metabolic pathways (e.g., carbohydrate metabolism, dormancy, and sporulation) following soil sterilization. Can these shifts be directly linked to changes in plant growth parameters?

That is a good point, but given the diversity and complexity of these metabolic pathways, a more detailed study of pathways and potential microbe-plant interactions would be required, which is beyond the scope of the current paper.

The study uses one-way ANOVA for comparing growth parameters. Were assumptions of normality and homogeneity of variance tested before applying ANOVA?

We did not test for homogeneity of variance.

The findings suggest that seed-associated fungi (Alternaria, Epicoccum) preferentially colonized sterile soil but not non-sterile soil. Could this be due to competitive exclusion in non-sterile soil, and was any addit

---

## [Decision Letter · Decision Letter 1]

4 Jul 2025

Dear Dr. Kyndt,

Thank you for submitting your manuscript to PLOS ONE. After careful consideration, we feel that it has merit but does not fully meet PLOS ONE’s publication criteria as it currently stands. Therefore, we invite you to submit a revised version of the manuscript that addresses the points raised during the review process.

We look forward to receiving your revised manuscript.

Kind regards,

Rajesh Singh Rathore, Ph.D

Academic Editor

PLOS ONE

Additional Editor Comments:

1. Material and Methods Section: Please provide proper citations for the materials, methods, instruments, and tools used in this study. For example, the tools used to measure temperature, humidity, and light levels should be cited appropriately. This will allow readers to better understand the methodology and replicate the study if needed.

2. Sample Collection Depth: For the soil collection process, please specify the depth at which the samples were collected. This detail is important for understanding the representativeness and the potential variability of the soil microbial communities across different soil layers.

3. EPA Method: In the material and methods section, please mention the specific Environmental Protection Agency (EPA) method or protocol followed for soil sample collection and analysis. This will ensure the method’s credibility and alignment with established standards for environmental and microbiological studies.

4. Seed Germination Measurement: How was seed germination measured in this study? Please specify the method used and provide the relevant reference for this procedure. Additionally, please elaborate on why the sterile soil showed a higher number of germinations compared to non-sterile soil. This is a crucial aspect to help readers understand the environmental conditions that influenced germination rates.

5. WGS Output and NMDS Analysis: In the results section, it is mentioned that WGS output was analyzed by selecting all reads aligned to the domain Bacteria, followed by NMDS analysis at the taxonomic level of genus. Could you please elaborate on this process? Specifically, how were the reads selected, and what was the rationale for performing NMDS analysis at the genus level? Additional details will help readers understand the methodology and the choices made during the analysis.

6. Figure 4 Clarifications: In Figure 4, the comparison between "sterile not planted" vs "sterile planted" shows higher bacterial and fungal diversity. Can you provide more details about this observation? Specifically, what could explain the higher diversity in the planted sterile soil?

7. The legend refers to Figure 4A and 4B, but these sub-figures are not included in the manuscript. Please check all figures and legends to ensure consistency and clarity.

8. The "soil metabolic pathways" shown in Figure 4 appear to be different from the figure referenced in the text. Kindly review and revise the figures and legends accordingly.

9. Figure Creation Process: Please specify in the methods section how the figures were created. Include the source of the input files, the file types used, and the tools/software utilized to generate the figures. For instance, it is unclear how Figure 5 was generated and how the relative abundance of soil metabolic pathway subsystems was calculated. Please clarify this process.

10. Raw Data Submission: Kindly provide the raw data (Whole genome, 16S, ITS) used in this study in supplementary material. Additionally, submit these data to NCBI and provide the accession numbers for these datasets.

11. Figure 6 Clarification: In Figure 6, why were only ~30 phyla identified for bacteria and 7 for fungi at the phylum level? Please provide an explanation for this observation.

12. Terminology Adjustment: Please modify the terminology from "whole genome" to "bacterial whole genome" or simply "bacteria" to provide a clearer indication of the focus on bacterial and fungal studies in your manuscript.

13. Basidiomycota and Actinobacteria Observations (Fig. 7):In Figure 7, it is noted that Basidiomycota were present in week 4 and week 7, but not at week 0, and the same observation is made for Actinobacteria. Could you please explain the reason behind this temporal variation? Additional details on this pattern will help readers understand the ecological implications.

14. Alpha and Beta Diversity: Please provide the alpha and beta diversity metrics, both week wise and treatment-wise, to give a clearer understanding of the differences between the samples over time and under different conditions. This will help in interpreting the microbial diversity patterns in the study.

Reviewers' comments:

Reviewer's Responses to Questions

**Comments to the Author**

Reviewer #1: All comments have been addressed

Reviewer #3: All comments have been addressed

2. Is the manuscript technically sound, and do the data support the conclusions?

Reviewer #1: Yes

Reviewer #3: Partly

3. Has the statistical analysis been performed appropriately and rigorously?

Reviewer #1: Yes

Reviewer #3: Yes

4. Have the authors made all data underlying the findings in their manuscript fully available?

Reviewer #1: Yes

Reviewer #3: No

5. Is the manuscript presented in an intelligible fashion and written in standard English?

Reviewer #1: No

Reviewer #3: Yes

Reviewer #1: The manuscript titled here is “Soil microbiome perturbation impedes growth of Bouteloua curtipendula and increases relative abundance of soil microbial pathogens”.

The authors have addressed the comments from the previous reviewers. However, some typographical and formatting issues still remain.

• For example, the species name B. curtipendula is not italicized in the heading of the Results section, and appears underlined elsewhere. Please ensure consistent and proper formatting of scientific names throughout the manuscript.

• Figure 1A: The proportions on the Y-axis appear to exceed the scale.

• I recommend a thorough proofreading of the entire manuscript to correct remaining typographical and formatting errors.

Reviewer #3: There are some critical issues that must be addressed before the manuscript can be considered for publication.

1. Experimental Design and Methodological Concerns

Lack of soil characterization: The study relies heavily on interpreting microbial community shifts, yet there is no physical or chemical analysis of soil properties beyond pH.

2. Language, Grammar, and Writing Style

Several long, multi-clause sentences would benefit from being split to improve clarity and reduce reader fatigue. There is occasional verb tense inconsistency, particularly between past and present tense in the Methods and Discussion sections.

Some stylistic redundancy exists in the Discussion, where results are repeated rather than critically analyzed.

3. Formatting and Editorial Corrections

Capitalization: Please correct all instances of “illumina” to “Illumina”, as this is a proper noun referring to a company and sequencing platform.

Figure Captions: Ensure all figure captions are written as complete sentences and include correct punctuation. For example:

Use proper spacing in statistical expressions (e.g., n = 6, not n=6).

**Do you want your identity to be public for this peer review?** For information about this choice, including consent withdrawal, please see our Privacy Policy

Reviewer #1: No

Reviewer #3: No

---

## [Author Response · Author response to Decision Letter 2]

22 Jul 2025

PONE-D-24-43622R1

Soil microbiome perturbation impedes growth of Bouteloua curtipendula and increases relative abundance of soil microbial pathogens

Dear Editor,

We appreciate the reviewers and your comments and have provided our rebuttal to each of the points below. In addition, we provided a marked-up document that highlights each of the changes made.

Additional Editor Comments:

1. Material and Methods Section: Please provide proper citations for the materials, methods, instruments, and tools used in this study. For example, the tools used to measure temperature, humidity, and light levels should be cited appropriately. This will allow readers to better understand the methodology and replicate the study if needed.

Temperature, humidity and light levels were controlled using the Greenhouse Wadsworth control systems using aspirated temperature/humidity sensors and PAR light sensors. The specific aspirator sensor model number has been added to the manuscript.

2. Sample Collection Depth: For the soil collection process, please specify the depth at which the samples were collected. This detail is important for understanding the representativeness and the potential variability of the soil microbial communities across different soil layers.

We collected the entirety of the soil. For these experiments, seeds were sown in small plug trays. At collection points, the plant material was removed and the entire plug of remaining soil was homogenized and sampled. This information is now added to the Mat and Meth section.

3. EPA Method: In the material and methods section, please mention the specific Environmental Protection Agency (EPA) method or protocol followed for soil sample collection and analysis. This will ensure the method’s credibility and alignment with established standards for environmental and microbiological studies.

We did not follow a specific EPA method. The available EPA methods are specifically designed for field soil measurements and not for greenhouse plant/soil sampling. The protocol steps are explained in detail in the Mat&Meth section and should allow for reproducibility by the readers.

4. Seed Germination Measurement: How was seed germination measured in this study? Please specify the method used and provide the relevant reference for this procedure. Additionally, please elaborate on why the sterile soil showed a higher number of germinations compared to non-sterile soil. This is a crucial aspect to help readers understand the environmental conditions that influenced germination rates.

In order to allow longitudinal study without disturbing the soil between collection points and to pair-match soil samples with plant growth data, germination was inferred as visible cotyledon. This is now added to the Mat & Meth section of the manuscript. Although more sensitive or specific measures for germination could be useful, they lack the convenience and capability of analyzing undisturbed samples.

5. WGS Output and NMDS Analysis: In the results section, it is mentioned that WGS output was analyzed by selecting all reads aligned to the domain Bacteria, followed by NMDS analysis at the taxonomic level of genus. Could you please elaborate on this process? Specifically, how were the reads selected, and what was the rationale for performing NMDS analysis at the genus level? Additional details will help readers understand the methodology and the choices made during the analysis.

The genus level is the standard for microbiome comparisons. None of the current methods allow for enough resolution at the species level. We don’t have enough resolution at the species level to have confidence in the species metric, so the genus level NMDS analysis must suffice. The species-level NMDS could over or underestimate microbial community differences between samples due to misidentifying particular reads as a given species.

6. Figure 4 Clarifications: In Figure 4, the comparison between "sterile not planted" vs "sterile planted" shows higher bacterial and fungal diversity. Can you provide more details about this observation? Specifically, what could explain the higher diversity in the planted sterile soil?

We believe this was already discussed (this relates to Figure 3, not Figure 4) and the explanation is that the plants likely provided additional niches to support new bacterial and fungal taxa. There is also an increase in potential plant pathogens following the introduction of plants to the sterilized soil. We have briefly expanded on this with the Figure 3 diversity figure in the Result section.

7. The legend refers to Figure 4A and 4B, but these sub-figures are not included in the manuscript. Please check all figures and legends to ensure consistency and clarity.

The line item in the text referring to 4A and 4B has been corrected. This referred to a previous version of the figure.

8. The "soil metabolic pathways" shown in Figure 4 appear to be different from the figure referenced in the text. Kindly review and revise the figures and legends accordingly.

The names of the metabolic pathways in the figure 4 and in the text describing this section (lines 353-360) are identical, and all are present in the figure. We are not sure where a revision might be needed.

9. Figure Creation Process: Please specify in the methods section how the figures were created. Include the source of the input files, the file types used, and the tools/software utilized to generate the figures. For instance, it is unclear how Figure 5 was generated and how the relative abundance of soil metabolic pathway subsystems was calculated. Please clarify this process.

All the figures were generated from raw Illumina sequence processing as described in the materials and methods. Delim .txt files of processed sequence data were analyzed in R, as described in the materials and methods. The relative abundance was calculated simply as the proportion (or percentage) of reads identified as a particular taxonomic unit out of all of the total reads. Or, in the case of soil metabolic pathway subsystems, the relative abundance is the percentage of reads for a particular subsystem out of all the reads analyzed. All analyses were performed in R and all graphs were generated in R using ggplot2. This information is included in the Mat & Meth, however we could provide the specific R scripts if deemed necessary (maybe as supplemental?). Including the specific, basic R scripts within the manuscripts seems like it would be outside the scope of the paper.

10. Raw Data Submission: Kindly provide the raw data (Whole genome, 16S, ITS) used in this study in supplementary material. Additionally, submit these data to NCBI and provide the accession numbers for these datasets.

The data was already submitted to NCBI and the accession numbers are already in the manuscript: (lines 205-209) “Both the WGS metagenomic and targeted fungal sequencing datasets were deposited into NCBI Genbank under BioProject PRJNA1163419 and the WGS data are accessible with the following SRA numbers: SRR30751922-SRR30751939 and SRR30754582-SRR30754613. The targeted fungal ITS sequencing files are available with the following SRA numbers: SRR30786940-SRR30786954”

11. Figure 6 Clarification: In Figure 6, why were only ~30 phyla identified for bacteria and 7 for fungi at the phylum level? Please provide an explanation for this observation.

The threshold of phyla greater than 2% of the rarefied reads is already included in the figure description. The phyla below the threshold were pooled into ‘other’ as described in the legend.

12. Terminology Adjustment: Please modify the terminology from "whole genome" to "bacterial whole genome" or simply "bacteria" to provide a clearer indication of the focus on bacterial and fungal studies in your manuscript.

The whole genome sequencing approach was not limited to bacteria only and therefore changing the terminology would be misleading. In the previous revision it was clarified why we did additional fungal ITS sequencing in addition to the WGS.

13. Basidiomycota and Actinobacteria Observations (Fig. 7):In Figure 7, it is noted that Basidiomycota were present in week 4 and week 7, but not at week 0, and the same observation is made for Actinobacteria. Could you please explain the reason behind this temporal variation? Additional details on this pattern will help readers understand the ecological implications.

We have now expanded on this observation in the Discussion section. The lack of these at week 0 is clearly due to the sterilization. However as we indicated, the biochemical reason for a possible inhibitory effect on Actinobacteria and stimulatory effect of Basidiomycota due to planting will need further detailed follow up studies.

14. Alpha and Beta Diversity: Please provide the alpha and beta diversity metrics, both week wise and treatment-wise, to give a clearer understanding of the differences between the samples over time and under different conditions. This will help in interpreting the microbial diversity patterns in the study.

The diversity information is already provided (figure 3 and figure 2), and we described how Shannon and beta diversity were calculated in the Mat&Meth.

Reviewers' comments:

Reviewer's Responses to Questions

Comments to the Author

1. If the authors have adequately addressed your comments raised in a previous round of review and you feel that this manuscript is now acceptable for publication, you may indicate that here to bypass the “Comments to the Author” section, enter your conflict of interest statement in the “Confidential to Editor” section, and submit your "Accept" recommendation.

Reviewer #1: All comments have been addressed

Reviewer #3: All comments have been addressed

2. Is the manuscript technically sound, and do the data support the conclusions?

Reviewer #1: Yes

Reviewer #3: Partly

3. Has the statistical analysis been performed appropriately and rigorously?

Reviewer #1: Yes

Reviewer #3: Yes

4. Have the authors made all data underlying the findings in their manuscript fully available?

Reviewer #1: Yes

Reviewer #3: No

5. Is the manuscript presented in an intelligible fashion and written in standard English?

Reviewer #1: No

Reviewer #3: Yes

6. Review Comments to the Author

Reviewer #1: The manuscript titled here is “Soil microbiome perturbation impedes growth of Bouteloua curtipendula and increases relative abundance of soil microbial pathogens”.

The authors have addressed the comments from the previous reviewers. However, some typographical and formatting issues still remain.

• For example, the species name B. curtipendula is not italicized in the heading of the Results section, and appears underlined elsewhere. Please ensure consistent and proper formatting of scientific names throughout the manuscript.

The species was not italicized in the result section headers because those were italicized themselves and standard nomenclature would invert italicized words in such instances. Nevertheless, we italicized everything for clarity. We did correct the underlined species name in the Result section.

• Figure 1A: The proportions on the Y-axis appear to exceed the scale.

The bars reach the 1.0 value with is the top of the scale, but do not exceed it (there would not be a value greater than 1 in the proportion scale).

• I recommend a thorough proofreading of the entire manuscript to correct remaining typographical and formatting errors.

This has been performed by two of the authors and errors have been corrected where needed.

Reviewer #3: There are some critical issues that must be addressed before the manuscript can be considered for publication.

1. Experimental Design and Methodological Concerns

Lack of soil characterization: The study relies heavily on interpreting microbial community shifts, yet there is no physical or chemical analysis of soil properties beyond pH.

As mentioned in the first review round, we did not test for chemical analysis in these experiments. All experiments were performed with the same starting soil and our main focus was measuring the microbial community changes with planting and sterilization. Some of these factors could have possibly impact the initial growth (of both microbes and plants). But if so, the plant growth rates are restored after microbes reestablish. If the sterilization changes additional properties of the soil, the plant growth inhibition of this change is ameliorated by the recolonization of bacteria/fungi. We have already expanded on this briefly in the discussion section.

2. Language, Grammar, and Writing Style

Several long, multi-clause sentences would benefit from being split to improve clarity and reduce reader fatigue. There is occasional verb tense inconsistency, particularly between past and present tense in the Methods and Discussion sections.

Some stylistic redundancy exists in the Discussion, where results are repeated rather than critically analyzed.

The verb tense inconsistencies in the Math and Methods have been corrected (at least two occasions were found).

Due to the complex and multi-level comparisons of this study, we felt the need to repeat some of the results in some sections of the discussion for clarity to the reader. There are various sections where we discuss planted vs not planted, sterile vs not sterile, bacteria vs fungi, etc, and repeating a brief summary of the results seem necessary to keep clarity for the readers. We did change/expanded some of the discussion sections to put the emphasis more on analysis (or future studies) and rearranged small sections and broke up some of the discussion section, which will hopefully help the readers.

3. Formatting and Editorial Corrections

Capitalization: Please correct all instances of “illumina” to “Illumina”, as this is a proper noun referring to a company and sequencing platform.

The term Illumina occurs 14 times in the manuscript and is capitalized each time. However, in two of the figure legends, the term was autocorrected by Word to ‘illumine’, which has now been corrected to “Illumina”.

Figure Captions: Ensure all figure captions are written as complete sentences and include correct punctuation. For example:

Use proper spacing in statistical expressions (e.g., n = 6, not n=6).

Figure legends and statistical expressions have been updated.

7. PLOS authors have the option to publish the peer review history of their article (what does this mean?). If published, this will include your full peer review and any attached files.

Do you want your identity to be public for this peer review? For information about this choice, including conse

---

## [Decision Letter · Decision Letter 2]

31 Aug 2025

Dear Dr. John Kyndt ,

We look forward to receiving your revised manuscript.

Kind regards,

Rajesh Singh Rathore, Ph.D

Academic Editor

PLOS ONE

Journal Requirements:

Additional Editor Comments (if provided):

Dear Dr. Kyndt,

Thank you for the opportunity to review your manuscript, "Soil Microbiome Perturbation Impedes Growth of Bouteloua curtipendula and Increases Relative Abundance of Soil Microbial Pathogens." The study is valuable and well-written. I have a few minor suggestions:

1. Please provide proper research article citations for the methodologies described, such as material methods, plant growth measurements, soil pH measurements, LB agar plating, DNA extraction, next-generation sequencing, and data analysis (e.g., MG-RAST using DESeq, etc.).

2. Please ensure that appropriate citations are provided for the instruments used in this study, including QuBit and NanoDrop.

3. The authors mentioned performing whole genome sequencing (WGS) for bacteria and then separately using ITS primers. This needs clarification: Does the WGS approach imply shotgun metagenomics sequencing?

4. Could the authors clarify whether microbial cell viability tests were conducted for both sterile and non-sterile samples? Amplicon primers were synthesized by Sigma Aldrich, what does it means?

5. Please provide proper reference for your statement -

6. While whole genome sequencing provides a comprehensive view of the microbiome, including functional genes and potential for strain-level resolution, however when using a WGS approach, the fungal species are often underrepresented (due to the abundance of bacteria) and therefore a more targeted approach is beneficial.

7. How does the variation in sequencing depth (139,594 to 9,540,606 reads) impact microbial diversity representation, particularly for low-abundance species?

8. How does sequencing depth influence community profiling and functional gene prediction?

9. What strategies can mitigate fungal underrepresentation in WGS due to bacterial abundance?

10. Does the low fungal representation in WGS affect the interpretation of soil microbiomes and plant interactions?

11. What biases might the Illumina Nextera DNA Flex Library Prep kit introduce in microbial taxonomic representation?

12. How could these biases affect community structure and functional profile interpretation?

13. What quality and concentration thresholds were used in Qubit to assess library quality?

14. How do these thresholds impact sequencing result accuracy and reproducibility?

15. How does Illumina MiniSeq's 2x150 bp read length affect strain-level resolution and functional annotation, particularly for complex soil microbiomes?

16. How does using the ITS1 region (18S to 5.8S rRNA genes) impact fungal taxonomic resolution in soil metagenomics?

17. Are there alternative markers that could improve fungal diversity analysis?

18. What measures can be taken to prevent or account for sample failures in sequencing and how do they affect result robustness?

19. How do different sequencing cassette configurations (High-output vs. Mid-output) affect data quality and interpretation?

20. How was data quality (Q30) ensured and what filtering criteria were used for consistency across the dataset?

21. What statistical methods assess the impact of sequencing variability (e.g., read numbers, data volume) on microbiome composition and functional interpretation?

22. Please include a section detailing the research contributions of each author to clarify their specific roles in the study.

23. Kindly provide a Declaration of Competing Interests to ensure transparency regarding any potential conflicts of interest.

24. The legend for Figure 3 needs to be rewritten. It is currently unclear which diversity measure (alpha or beta diversity) is being discussed. Could you please specify and provide clarification?

25. To enhance reproducibility, please provide the scripts or detailed procedure used for processing data with MG-RAST (or any other GUI tool). This should be included as a supplementary file so that other users can replicate your methods.

26. The legends for Figures 5, 8, and 9 require revision. The current descriptions are unclear. Additionally, please describe the figures as subfigures within the legend for clarity.

27. Please correct writing style of phyla. Order, and genus in whole manuscript–

a) Phylum (or Phyla in plural) names are always capitalized and italicized or underlined. Example: Actinobacteria, Bacteroidetes

b) Order names are capitalized but not italicized. Example: Actinomycetales, Bacteroidales

c) Genus names are capitalized and italicized (or underlined if handwritten). Example: Streptomyces, Bacteroides

d) Species names are not capitalized but are italicized (or underlined if handwritten). Example: Streptomyces coelicolor, Bacteroides fragilis

Suggested Reference-

Caporaso JG, Lauber CL, Walters WA, Berg-Lyons D, Huntley J, Fierer N, et al. Ultrahigh-throughput microbial community analysis on the Illumina HiSeq and MiSeq platforms. The ISME journal. 2012;6(8):1621-4.

Pathak A, Jaswal R, Xu X, White JR, Edwards III B, Hunt J, et al. Characterization of bacterial and fungal assemblages from historically contaminated metalliferous soils using metagenomics coupled with diffusion chambers and microbial traps. Frontiers in Microbiology. 2020;11:1024

Gendy S, Chauhan A, Agarwal M, Pathak A, Rathore RS, Jaswal R. Is long-term heavy metal exposure driving carriage of antibiotic resistance in environmental opportunistic pathogens: a comprehensive phenomic and genomic assessment using Serratia sp. SRS-8-S2018. Frontiers in Microbiology. 2020;11:1923.

Caporaso JG, Lauber CL, Walters WA, Berg-Lyons D, Lozupone CA, Turnbaugh PJ, et al. Global patterns of 16S rRNA diversity at a depth of millions of sequences per sample. Proceedings of the national academy of sciences. 2011;108(supplement_1):4516-22.

Clagnan E, Costanzo M, Visca A, Di Gregorio L, Tabacchioni S, Colantoni E, et al. Culturomics-and metagenomics-based insights into the soil microbiome preservation and application for sustainable agriculture. Frontiers in Microbiology. 2024;15:1473666.

Caporaso JG, Lauber CL, Walters WA, Berg-Lyons D, Huntley J, Fierer N, et al. Ultrahigh-throughput microbial community analysis on the Illumina HiSeq and MiSeq platforms. The ISME journal. 2012;6(8):1621-4.

Yang C, Mai J, Cao X, Burberry A, Cominelli F, Zhang L. ggpicrust2: an R package for PICRUSt2 predicted functional profile analysis and visualization. Bioinformatics. 2023;39(8):btad470.

Reviewers' comments:

Reviewer's Responses to Questions

**Comments to the Author**

Reviewer #1: All comments have been addressed

2. Is the manuscript technically sound, and do the data support the conclusions?

Reviewer #1: Yes

3. Has the statistical analysis been performed appropriately and rigorously?

Reviewer #1: Yes

4. Have the authors made all data underlying the findings in their manuscript fully available?

Reviewer #1: Yes

5. Is the manuscript presented in an intelligible fashion and written in standard English?

Reviewer #1: Yes

Reviewer #1: The authors have adequately addressed all reviewer comments and concerns in the revised manuscript. As such, the paper is now acceptable for publication.

**Do you want your identity to be public for this peer review?** For information about this choice, including consent withdrawal, please see our Privacy Policy

Reviewer #1: No

---

## [Author Response · Author response to Decision Letter 3]

18 Sep 2025

Dear Editor,

We are glad that all the reviewer comments were satisfied and appreciate your additional comments for improvement. Below we provide our rebuttal to each of the points and pointed out which sections were further clarified in the text. In addition, we provided a marked-up document that highlights each of the changes made.

Additional Editor Comments

Dear Dr. Kyndt,

Thank you for the opportunity to review your manuscript, "Soil Microbiome Perturbation Impedes Growth of Bouteloua curtipendula and Increases Relative Abundance of Soil Microbial Pathogens." The study is valuable and well-written. I have a few minor suggestions:

1. Please provide proper research article citations for the methodologies described, such as material methods, plant growth measurements, soil pH measurements, LB agar plating, DNA extraction, next-generation sequencing, and data analysis (e.g., MG-RAST using DESeq, etc.).

The plant growth and soil pH measurement were not based on previous research articles and the methods and consumables used are described in this manuscript in a manner that should be reproducible by the reader. DNA extraction and NGS sequencing was performed using commercial kits and an in-house Illumina sequencer, and we described this in detail in the Mat and Meth. A reference for DESeq has been added in addition to the MG-RAST references that were already provided.

2. Please ensure that appropriate citations are provided for the instruments used in this study, including QuBit and NanoDrop.

The company and model information was added for both of these.

3. The authors mentioned performing whole genome sequencing (WGS) for bacteria and then separately using ITS primers. This needs clarification: Does the WGS approach imply shotgun metagenomics sequencing?

Yes, it is now added that WGS is a shotgun sequencing approach.

4. Could the authors clarify whether microbial cell viability tests were conducted for both sterile and non-sterile samples? Amplicon primers were synthesized by Sigma Aldrich, what does it means?

Viability tests were only performed to the sterile soil samples to identify any possible residual viable cultures (as described in the paper). Viability tests on all other samples would not be very informative as only a small percentage of the soil organisms would actually be able to be tested or cultivate under laboratory conditions, and hence this would not contribute anything to this study. Sigma Aldrich is a well-known chemical company that synthesizes custom primer orders. They are now part of the larger Merck conglomerate. We indicated the Merck company, state and country references to this statement for clarity.

5. Please provide proper reference for your statement – which statement? I assume this refers to point 6 below…

6. While whole genome sequencing provides a comprehensive view of the microbiome, including functional genes and potential for strain-level resolution, however when using a WGS approach, the fungal species are often underrepresented (due to the abundance of bacteria) and therefore a more targeted approach is beneficial.

Two references were added here, and it was clarified that the lack of curated fungal sequences in the WGS databases are also a contributing factor.

7. How does the variation in sequencing depth (139,594 to 9,540,606 reads) impact microbial diversity representation, particularly for low-abundance species?

This is why rarefaction and normalization of the data was performed in the analysis (as indicated in the Mat and Meth section.) This point was also previously addressed in the reviewer comments.

8. How does sequencing depth influence community profiling and functional gene prediction? This can certainly have an impact, however all of our samples passed the MG-RAST initial QC that takes sequencing depth into consideration.

9. What strategies can mitigate fungal underrepresentation in WGS due to bacterial abundance? This is why we performed the additional targeted ITS sequencing on top of the WGS approach. This is explained in the text (and in comment 6 above).

10. Does the low fungal representation in WGS affect the interpretation of soil microbiomes and plant interactions?

This is why we performed the additional targeted ITS sequencing on top of the WGS approach and the fungal results were discussed in conjunction with the bacterial results. Since we performed the targeted ITS fungal sequencing we can describe and compare the fungal impact. The ‘low representation’ is only a sequencing artifact and by taking the ITS fungal data into account we have provided a more realistic interpretation of soil microbes and plant interactions.

11. What biases might the Illumina Nextera DNA Flex Library Prep kit introduce in microbial taxonomic representation? The Illumina Nextera DNA Flex Library Prep kit was specifically designed to minimize the GC content and coverage biases that plagued its predecessor, the Nextera XT kit. However, no library preparation method is entirely free of bias. The most significant potential bias stems from the DNA extraction process, but the Flex kit itself can still introduce minor, though generally not substantial biases in taxonomic representation.

12. How could these biases affect community structure and functional profile interpretation?

Multiple studies comparing Flex to other library prep methods found it introduces very little GC-related bias and produces a more uniform coverage across genomes with varying GC content. These are accepted standards and library preps in microbial genome sequencing and are the best we can currently do for analyzing community structure and functional profiling.

13. What quality and concentration thresholds were used in Qubit to assess library quality?

The purity and concentration levels for Qubit are added to the paper. All library samples had A260/280 ratios above 1.8 and all concentrations were above 10nM (which is the starting pool concentration).

14. How do these thresholds impact sequencing result accuracy and reproducibility?

All samples were well above these threshold.

15. How does Illumina MiniSeq's 2x150 bp read length affect strain-level resolution and functional annotation, particularly for complex soil microbiomes?

Any short read sequencing (and even long read) sequencing technique has issues with strain-level resolution in complex microbiomes, which is why we (and most other microbiome analysis papers) limit their comparison to genus (and sometimes species) level comparisons. Illumina’s 2x150 bp read length is a standard for microbiome level comparisons and well suited for genus level comparisons (not strain) and functional annotation. We are not performing strain level comparisons in this paper.

16. How does using the ITS1 region (18S to 5.8S rRNA genes) impact fungal taxonomic resolution in soil metagenomics? There is ongoing debate on which ITS region (ITS1/ITS2) is best for fungal genetic analysis, but both are widely used for metagenomic analysis. We chose the ITS1 region as per the demonstrated Illumina Fungal Metagenomic Sequencing Protocol (as indicated in the text).

17. Are there alternative markers that could improve fungal diversity analysis?

See answer to 16 above. The ITS regions are the only fungal markers that currently have sufficient database representation.

18. What measures can be taken to prevent or account for sample failures in sequencing and how do they affect result robustness?

Is this a general question? Every experiment will have to attempt to prevent sample failure and has a need for replicas. We have 6 biological replicas for each of the sampling conditions, which should provide sufficient robustness in the study. This is already described in the paper.

19. How do different sequencing cassette configurations (High-output vs. Mid-output) affect data quality and interpretation? The different cassettes do not impact data quality, the High-output simply allows for more samples per run, provided that there is enough sequencing depth for each sample.

20. How was data quality (Q30) ensured and what filtering criteria were used for consistency across the dataset?

The low quality data (<Q30) is already filter out by the sequencer data processing, however we performed an additional filtering of low qualtity scored reads using the FastQ Toolkit in BaseSpace (Illumina), as indicated under the Data Analysis Mat&Meth section.

21. What statistical methods assess the impact of sequencing variability (e.g., read numbers, data volume) on microbiome composition and functional interpretation?

As indicated in the text, rarefaction and normalization of the reads was performed either in R (scripts are now provided as supplemental info) or in MG-RAST itself.

22. Please include a section detailing the research contributions of each author to clarify their specific roles in the study.

A section with author contributions is now added after the Acknowledgements

23. Kindly provide a Declaration of Competing Interests to ensure transparency regarding any potential conflicts of interest.

A declaration of no conflicts of interest is now added after the Acknowldgements.

24. The legend for Figure 3 needs to be rewritten. It is currently unclear which diversity measure (alpha or beta diversity) is being discussed. Could you please specify and provide clarification?

It is now indicated in Figure 3 that alpha diversity (Shannon diversity index) is being discussed.

25. To enhance reproducibility, please provide the scripts or detailed procedure used for processing data with MG-RAST (or any other GUI tool). This should be included as a supplementary file so that other users can replicate your methods.

MG-RAST is a public service and scripts have been published before (and referenced in our paper). The additional R-scripts that we used and described in the manuscript are now submitted as supplemental information.

26. The legends for Figures 5, 8, and 9 require revision. The current descriptions are unclear. Additionally, please describe the figures as subfigures within the legend for clarity.

We have now restructured the descriptions for these figures to indicate more clearly what the subfigures are and how they should be interpreted, at the beginning of each description. We feel the methods and descriptions are clear as is but could address specific confusions if the editor has them.

27. Please correct writing style of phyla. Order, and genus in whole manuscript–

a) Phylum (or Phyla in plural) names are always capitalized and italicized or underlined. Example: Actinobacteria, Bacteroidetes

b) Order names are capitalized but not italicized. Example: Actinomycetales, Bacteroidales

c) Genus names are capitalized and italicized (or underlined if handwritten). Example: Streptomyces, Bacteroides

d) Species names are not capitalized but are italicized (or underlined if handwritten). Example: Streptomyces coelicolor, Bacteroides fragilis

The manuscript has been updated to reflect these recommendations.

Suggested Reference-

Caporaso JG, Lauber CL, Walters WA, Berg-Lyons D, Huntley J, Fierer N, et al. Ultrahigh-throughput microbial community analysis on the Illumina HiSeq and MiSeq platforms. The ISME journal. 2012;6(8):1621-4.

We are not sure why this paper is recommended. The paper discusses alternative Illumina platforms to the MiniSeq we used. Our MiniSeq instrument uses similar chemistry to the Nextseq technology, which is newer sequencing chemistry than the older Hiseq and MiSeq platforms. The newer platforms allow for faster sequencing and higher accuracy.

Pathak A, Jaswal R, Xu X, White JR, Edwards III B, Hunt J, et al. Characterization of bacterial and fungal assemblages from historically contaminated metalliferous soils using metagenomics coupled with diffusion chambers and microbial traps. Frontiers in Microbiology. 2020;11:1024

Although this is an interesting paper describing the bacterial and fungal communities in metal contaminated soils (particularly mercury) and an interesting DC/MT technology, since we did not examine metal content in our soil (and don’t suspect high metal contamination), we are not sure how this study is particularly relevant to our current analysis.

Gendy S, Chauhan A, Agarwal M, Pathak A, Rathore RS, Jaswal R. Is long-term heavy metal exposure driving carriage of antibiotic resistance in environmental opportunistic pathogens: a comprehensive phenomic and genomic assessment using Serratia sp. SRS-8-S2018. Frontiers in Microbiology. 2020;11:1923.

Similar to the reference before, this is an interesting paper in reference to long-term contaminated heavy metal habitats and the spread of antibiotic resistance by specific Serratia species in such sites. However since we did not find substantial levels of Serratia and have no indication of long-term heavy metal contamination in our soil (or in native prairie soils), adding this paper would create an additional justification and discussion that is beyond the scope of this paper.

Caporaso JG, Lauber CL, Walters WA, Berg-Lyons D, Lozupone CA, Turnbaugh PJ, et al. Global patterns of 16S rRNA diversity at a depth of millions of sequences per sample. Proceedings of the national academy of sciences. 2011;108(supplement_1):4516-22.

We are not sure why this reference is recommended. We did not perform 16S metagenomics analysis for our study. We purposely performed WGS metagenomics and have much higher coverage per sample than what is discussed in this (older) paper.

Clagnan E, Costanzo M, Visca A, Di Gregorio L, Tabacchioni S, Colantoni E, et al. Culturomics-and metagenomics-based insights into the soil microbiome preservation and application for sustainable agriculture. Frontiers in Microbiology. 2024;15:1473666.

This is a good general paper about holistic approaches for sustainable agriculture. Similar approaches could be used as we suggested in our discussion of native plant establishment. We added this reference (and another related one) and expanded briefly on this in our final discussion.

Caporaso JG, Lauber CL, Walters WA, Berg-Lyons D, Huntley J, Fierer N, et al. Ultrahigh-throughput microbial community analysis on the Illumina HiSeq and MiSeq platforms. The ISME journal. 2012;6(8):1621-4.

This is the same reference as the first one. See comment above.

Yang C, Mai J, Cao X, Burberry A, Cominelli F, Zhang L. ggpicrust2: an R package for PICRUSt2 predicted functional profile analysis and visualization. Bioinformatics. 2023;39(8):btad470.

We are not sure why this reference is recommended. We did not use this specific R package in our analysis.

---

## [Decision Letter · Decision Letter 3]

9 Dec 2025

Soil microbiome perturbation impedes growth of Bouteloua curtipendula and increases relative abundance of soil microbial pathogens

PONE-D-24-43622R3

Dear Dr. John Kyndt,

We’re pleased to inform you that your manuscript has been judged scientifically suitable for publication and will be formally accepted for publication once it meets all outstanding technical requirements.

Kind regards,

Rajesh Singh Rathore, Ph.D

Academic Editor

PLOS One

Additional Editor Comments (optional):

Editor’s Note to Authors (Final Accepted Manuscript) -

In the accepted manuscript, please ensure that the following reviewer comments are clearly addressed:

“The authors have expanded the discussion as requested, addressing ecological mechanisms (differential recovery of bacteria and fungi, competitive exclusion, functional redundancy) and thoughtfully contextualizing their findings within real-world restoration scenarios.”

→ This has been satisfactorily incorporated into the Discussion section.

“The authors provide a pragmatic but not entirely conclusive response to the critique that autoclaving may alter soil abiotic properties. Their argument that ‘if changes occurred, microbial recolonization mitigated them’ is logical but partly circular. This is an inherent methodological constraint of sterilization studies. To strengthen your argument regarding sterilization effects, consider adding a brief acknowledgment in the Discussion.”

→ Please ensure that the Discussion contains a concise acknowledgment of the methodological limitation posed by autoclaving and its potential abiotic effects, clarifying how this constraint influences interpretation of the results.

Reviewers' comments:

Reviewer's Responses to Questions

**Comments to the Author**

Reviewer #1: All comments have been addressed

Reviewer #3: All comments have been addressed

2. Is the manuscript technically sound, and do the data support the conclusions?

Reviewer #1: Yes

Reviewer #3: Yes

3. Has the statistical analysis been performed appropriately and rigorously?

Reviewer #1: Yes

Reviewer #3: Yes

4. Have the authors made all data underlying the findings in their manuscript fully available?

Reviewer #1: Yes

Reviewer #3: Yes

5. Is the manuscript presented in an intelligible fashion and written in standard English?

Reviewer #1: Yes

Reviewer #3: Yes

Reviewer #1: Thank you for providing the revised MS. All comments have been addressed now by the authors and can be accepted for the publication.

Reviewer #3: The authors have expanded the discussion as requested, addressing ecological mechanisms (differential recovery of bacteria and fungi, competitive exclusion, functional redundancy) and thoughtfully contextualizing their findings within real-world restoration scenarios.

he authors provide a pragmatic but not entirely conclusive response to the critique that autoclaving may alter soil abiotic properties. Their argument that "if changes occurred, microbial recolonization mitigated them" is logical but partly circular. This is an inherent methodological constraint of sterilization studies.To strengthen your argument regarding sterilization effects, consider adding a brief acknowledgment in the Discussion.

**Do you want your identity to be public for this peer review?** For information about this choice, including consent withdrawal, please see our Privacy Policy

Reviewer #1: No

Reviewer #3: No

---

## [Editor Report · Acceptance letter]

PONE-D-24-43622R3

PLOS One

Dear Dr. Kyndt,

I'm pleased to inform you that your manuscript has been deemed suitable for publication in PLOS One. Congratulations! Your manuscript is now being handed over to our production team.

Kind regards,

on behalf of

Dr. Rajesh Singh Rathore

Academic Editor

PLOS One